# A multiscale model of complex endothelial cell dynamics in early angiogenesis

**Daria Stepanova**[1,2]*, **Helen M. Byrne**[3], **Philip K. Maini**[3], **Tomás Alarcón**[1,2,4,5]

**1** Centre de Recerca Matemàtica, Bellaterra (Barcelona), Spain, **2** Departament de Matemàtiques, Universitat Autònoma de Barcelona, Bellaterra (Barcelona), Spain, **3** Wolfson Centre for Mathematical Biology, Mathematical Institute, University of Oxford, Oxford, UK, **4** Institució Catalana de Recerca i Estudis Avançats (ICREA), Barcelona, Spain, **5** Barcelona Graduate School of Mathematics (BGSMath), Barcelona, Spain

* dstepanova@crm.cat

**Data Availability Statement:** All relevant data are within the manuscript and its Supporting information files.

**Funding:** This work is supported by a grant of the Obra Social La Caixa Foundation on Collaborative

## Abstract

We introduce a hybrid two-dimensional multiscale model of angiogenesis, the process by which endothelial cells (ECs) migrate from a pre-existing vascular bed in response to local environmental cues and cell-cell interactions, to create a new vascular network. Recent experimental studies have highlighted a central role of cell rearrangements in the formation of angiogenic networks. Our model accounts for this phenomenon via the heterogeneous response of ECs to their microenvironment. These cell rearrangements, in turn, dynamically remodel the local environment. The model reproduces characteristic features of angiogenic sprouting that include branching, chemotactic sensitivity, the brush border effect, and cell mixing. These properties, rather than being hardwired into the model, emerge naturally from the gene expression patterns of individual cells. After calibrating and validating our model against experimental data, we use it to predict how the structure of the vascular network changes as the baseline gene expression levels of the VEGF-Delta-Notch pathway, and the composition of the extracellular environment, vary. In order to investigate the impact of cell rearrangements on the vascular network structure, we introduce the mixing measure, a scalar metric that quantifies cell mixing as the vascular network grows. We calculate the mixing measure for the simulated vascular networks generated by ECs of different lineages (wild type cells and mutant cells with impaired expression of a specific receptor). Our results show that the time evolution of the mixing measure is directly correlated to the generic features of the vascular branching pattern, thus, supporting the hypothesis that cell rearrangements play an essential role in sprouting angiogenesis. Furthermore, we predict that lower cell rearrangement leads to an imbalance between branching and sprout elongation. Since the computation of this statistic requires only individual cell trajectories, it can be computed for networks generated in biological experiments, making it a potential biomarker for pathological angiogenesis.

Mathematics awarded to the Centre de Recerca Matemàtica through a scholarship awarded to D.S. D.S. and T.A. have been partially funded by the CERCA Programme of the Generalitat de Catalunya. They also acknowledge MINECO (https://www.ciencia.gob.es/) for funding under grants MTM2015-71509-C2-1-R and RTI2018-098322-B-I00. D.S. and T.A. participate in project 2017SGR01735 which was awarded by AGAUR (https://agaur.gencat.cat/en/inici/index.html) but with no actual funding. The funders had no role in study design, data collection and analysis, decision to publish, or preparation of the manuscript.

**Competing interests:** The authors have declared that no competing interests exist.

## Author summary

Angiogenesis, the process by which new blood vessels are formed by sprouting from the pre-existing vascular bed, plays a key role in both physiological and pathological processes, including tumour growth. The structure of a growing vascular network is determined by the coordinated behaviour of endothelial cells in response to various signalling cues. Recent experimental studies have highlighted the importance of cell rearrangements as a driver for sprout elongation. However, the functional role of this phenomenon remains unclear. We formulate a new multiscale model of angiogenesis which, by accounting explicitly for the complex dynamics of endothelial cells within growing angiogenic sprouts, is able to reproduce generic features of angiogenic structures (branching, chemotactic sensitivity, cell mixing, etc.) as emergent properties of its dynamics. We validate our model against experimental data and then use it to quantify the phenomenon of cell mixing in vascular networks generated by endothelial cells of different lineages. Our results show that there is a direct correlation between the time evolution of cell mixing in a growing vascular network and its branching structure, thus paving the way for understanding the functional role of cell rearrangements in angiogenesis.

## Introduction

Angiogenesis, the process by which new blood vessels are generated from a pre-existing vascular network, has been extensively investigated in recent years [1–10]. However, understanding of the complex behaviour of endothelial cells (ECs) during angiogenesis remains incomplete. ECs adjust their gene expression profiles in response to a variety of extracellular cues, such as the structure and composition of the extracellular matrix (ECM), angiogenic factors such as vascular endothelial growth factor (VEGF), and changes in cell-cell interactions. Broadly, the extracellular stimuli act upon the VEGF-Delta-Notch signalling pathway, which drives ECs to acquire either a tip or stalk cell phenotype [11, 12]. Tip cells, with elevated expression of Delta ligand, VEGF receptor 2 (VEGFR2) and decreased expression of Notch receptor, are more active than their stalk cell counterparts [13]; they extend filopodia, release matrix metalloproteinases (MMPs) that degrade the ECM and, together with the pericytes that they recruit, secrete basal lamina components that stabilise growing vessels. By contrast, stalk cells, characterised by low expression of Delta and VEGFR2, and high Notch, migrate along the paths explored by tip cells, and proliferate to replenish the elongating sprout. EC phenotypes are not static and strongly depend on the local microenvironment (signalling cues, interactions with other ECs, among others) [2, 7]. Furthermore, ECs within sprouts exchange their relative positions via a phenomenon called *cell mixing* [2, 7]. Every time a cell rearrangement takes place, the cells' microenvironment changes. This, in turn, leads to recurrent (re-)establishment of phenotypes and variations in gene expression within the phenotype. As a consequence, a dynamic coupling between cell phenotypes and rearrangements is established. The functional role of cell mixing, and how it is affected by variations in the gene expression patterns of ECs, are unclear, although it is acknowledged that cell rearrangements greatly influence the pattern of the vascular network and its functionality [1, 7, 14, 15].

Angiogenic sprouting has been extensively studied from a theoretical modelling perspective in numerous physiological and pathological contexts, including tumour growth [10, 16–33]. Many models [18–20, 22, 24–28, 32] fall within the so-called snail-trail paradigm [31]. In this approach, EC phenotype acquisition is assumed to be irreversible; a subset of cells, associated with the tip phenotype and situated at the leading edge of growing sprouts, responds

chemotactically to gradients of signalling cues such as VEGF. The rest of the cells are assumed to be stalk cells which passively follow cells at the leading edge of the sprouts. Cell rearrangements are neglected and properties such as branching are hardwired via *ad hoc* rules, where the probability of branching depends on environmental factors [18–20, 24, 25, 28]. Furthermore, in the majority of models, chemotactic behaviour is coded into the model via an explicit bias of the migration probabilities towards regions of higher VEGF concentration [18–20, 24–28]. As a result, these models are unable to explain how variations in subcellular signalling, which determine cell phenotype, modify the structure of the growing network, or how specific mutations might lead to pathological networks. A more extensive review on the mathematical and computational models of angiogenesis can be found in [34–36].

Several recent computational models have started to address these issues [3, 37]. For example, Bentley et al. [3] studied the mechanisms underlying cell rearrangement behaviour during angiogenic sprouting. Using computational modelling and experimentation, they identified the Notch/VEGFR-regulated dynamics of EC adhesion as a key driver of EC rearrangements. Furthermore, simulation and quantitative image analysis indicated that abnormal phenotype synchronisation exists under pathologically high VEGF conditions, in retinopathy and tumour vessels [3]. However, due to the computational intensity of the model, only small scale simulations (a single vessel with 10 ECs) were performed. Boas and Merks [37] have also studied EC rearrangement within small vascular networks. They investigated how a cell at the leading position in a sprout can be overtaken by another cell, a phenomenon called *cell overtaking*. Their simulation results, obtained using the cellular Potts framework, suggest that cell overtaking occurs in an unregulated fashion, due to the stochastic motion of ECs. Moreover, their findings suggest that the role of the VEGF-Delta-Notch pathway in cell overtaking is not to select a cell with the highest expression of Delta ligand and VEGFR2, corresponding to the tip phenotype, that will shuffle up towards the sprout tip. Rather, cell overtaking is proposed to ensure that a cell that ends up occupying the leading position in a vessel, due to its spontaneous migration backward and forward within the sprout, acquires the tip phenotype. In their model, time-dependent ordinary differential equations (ODEs) are used to simulate the dynamics of subcellular signalling pathways within each cell. Cells are assigned a discrete phenotype, tip or stalk, and variations in their gene expression levels that might influence the behaviour of the cell were neglected.

Cell rearrangements have been proposed to be the main driver for early sprouting angiogenesis [1–3, 7]. In particular, reduced cell mixing leads to formation of pathological networks characterised by superimposed aberrant layers of vessels [1]. In order to characterise this phenomenon in a quantitative manner and determine its influence on the growth of vascular networks, we introduce a multiscale mathematical model of early angiogenesis (i.e. on a time scale of hours where cell birth/death are negligible). The model accounts for individual cell gene expression patterns associated with the VEGF-Delta-Notch signalling pathway that define two distinct cell phenotypes. Gene expression patterns are dynamically updated upon cell migration and in response to fluctuations in the cell microenvironment. Since cell behaviour has been shown to be heterogeneous and vary significantly depending on phenotype, cell-ECM interactions (ECM proteolysis, ECM remodelling, alignment of ECM fibrils) are assumed to be phenotype-dependent. In turn, the ECM configuration directly influences EC polarity and migration, thus changing the local environment of cells and providing coupling between the subcellular, cellular and tissue scales of the model.

One of our main results is that the model is capable of reproducing the typical behaviour of ECs; branching, chemotactic sensitivity and cell mixing are emergent properties of the model (instead of being hardwired into it) that arise as a result of cell-ECM interactions involving cells with dynamic subcellular gene expression. By calibrating and validating our model

against experimental data, we can investigate the role that abnormal subcellular signalling has on cell-matrix interactions, cell rearrangements and the general structure of the growing vascular network. Notably, we show how externally imposed changes in the extracellular environment (ECM density, VEGF concentration, VEGF gradient) and mutations in gene expression of ECs can alter the branching pattern of growing sprouts.

To quantify cell mixing within the vascular network, we introduce a *mixing measure*. This scalar metric is calculated for cells that are immediate neighbours at a given time moment, characterising how far they spread within the vascular network migrating during a fixed time interval. Our results suggest that in a formed network, temporal evolution of the mixing measure reaches a steady state that depends on the relative proportion of EC tip and stalk phenotypes. However, during the early stages of network formation, the time evolution of the mixing measure varies as the extracellular environment and cells' expression levels of genes such as VEGFR1 and VEGFR2 change. In particular, we show that networks created by mutant cells with impaired gene expression of VEGF receptors exhibit a delayed mixing compared to the networks formed by wild type cells with unaltered gene expression. The networks formed by these mutants demonstrate an imbalance between effective sprout elongation and branching. Therefore, our results confirm the crucial role of cell rearrangements in the formation of functional vascular networks in sprouting angiogenesis [1, 7].

The remainder of this paper is organised as follows. In the **Experimental motivation** section, we summarise the setups and results of several experimental studies which motivated the formulation of our multiscale model. The **Materials and methods** section contains a summary of the relevant biological information, a description of our multiscale model and metrics used to analyse vascular network evolution. In the **Results** section we compare our simulation results with data from Arima et al. [2] and Shamloo & Heilshorn [38]. These data were extracted from *in vitro* experiments, and enable us to define a set of basal parameter values for our model. Further model validation is performed against experimental results involving mutant cells, with modified expression of VEGF receptors, carried out by Jakobsson et al. [7]. Finally, we present results on EC mixing quantification and show how it relates to the different branching patterns of the growing vascular networks. We conclude by drawing together our findings and outlining possible avenues for future work in the **Discussion** section.

## Experimental motivation

The model we develop is motivated by *in vitro* experiments in which an aortic ring assay was embedded into a collagen matrix with a uniform VEGF concentration (0, 5 or 50 ng/ml) [2, 8]. Computational analysis of dynamic images, collected using time-lapse microscopy, revealed complex dynamical cell rearrangements within growing sprouts, a phenomenon termed *cell mixing*. The authors concluded that over short periods of time (e.g. 22.4 h averaged over all experiments in [2]) cell rearrangements are the main driver of sprout elongation. Interestingly, successful sprout growth was seen in a uniform distribution of VEGF across the substrate—sprout elongation velocity was observed to vary as the concentration of the external VEGF was changed, but no VEGF gradient was necessary for coordinated migration of ECs.

Dynamic cell rearrangements within elongating sprouts are a direct consequence of cells continuously updating their phenotype (i.e. adapting their gene expression pattern, depending on their environment [1, 3, 7]). Jakobsson et al. [7] identified the VEGF-Delta-Notch signalling pathway as the key pathway controlling this phenomenon. Using mutant cells heterozygous for VEGFR1 (VEGFR1$^{+/-}$) and VEGFR2 (VEGFR2$^{+/-}$) with halved (compared to wild-type (WT) cells) gene expression of the corresponding VEGF receptor, they investigated how differential levels of VEGF receptors affect the probability that a WT or mutant EC will occupy

the leading position in a growing sprout. Embryoid bodies (three-dimensional spherical aggregates of cells) derived from WT cells mixed with one of the populations of mutant cells (50% and 50%) were indistinguishable from those formed by WT cells only, however, the contribution of each cell line to the leading position differed. In particular, VEGFR1$^{+/-}$ (VEGFR2$^{+/-}$) cells demonstrated enhanced (reduced) competition for the leading cell position. The role of the VEGF-Delta-Notch signalling pathway in establishing competitive advantage was reinforced by experiments with the DAPT inhibitor, which abolishes Notch signalling in all EC lineages. Treatment with DAPT, although leading to hyper-sprouting, restored balance in competition for the leading cell position. Motivated by these results and our interest in studying cell rearrangements, we account for the VEGF-Delta-Notch signalling pathway of individual cells in our model.

The coordinated migration of ECs is a result of cell-ECM interactions [39–43]. Specifically, cell migration depends on the EC ability to degrade ECM proteins via proteolysis in order to form ECM-free vascular guidance tunnels for effective sprout elongation. This was confirmed by experimental results performed by Shamloo et al. [38]. Therein, ECs were cultured within a microfluidic device with a maintained gradient of VEGF concentration and the response of ECs to variations in ECM components, specifically collagen, was considered. Their results showed that there is a "sweet spot" of collagen density for the formation of angiogenic sprouts. In low collagen densities, ECs migrate freely into the ECM without forming sprouts; at intermediate concentrations, structures resembling angiogenic sprouts form; at high collagen densities, ECs are unable to migrate into the matrix significantly and form thick short protrusions. These experimental results motivated us to include cell-ECM interactions in our model.

Most *in vitro* experiments are performed on flat substrates in which the depth of the substrate can be considered negligible compared to its length and width [2, 8, 38, 40]. Thus, we formulate our model in a two-dimensional framework.

## Materials and methods

### Biological summary

Angiogenic sprouting is initiated when an active cytokine, such as VEGF, reaches the existing vasculature and activates quiescent ECs. Active ECs express proteases that degrade the basement membrane (BM) so that EC extravasation and migration can occur [44, 45]. *In vivo* and *in vitro*, coordinated migration is guided by local chemical and mechanical cues [42, 46, 47]. While VEGF-activated ECs at the front of the vascular network explore new space and anastomose, to form branching patterns, previously created vascular sprouts mature, form lumens and, with the help of murine cells (pericytes and smooth muscle cells), stabilise [48]. In our mathematical model, we focus on EC migration, and its regulation by local environmental (mechanical and chemical) cues. Due to the short time scales considered in the model, processes such as proliferation, vessel maturation and lumen formation are neglected.

VEGF-induced actin polymerisation and focal adhesion assembly result in substantial cytoskeletal remodelling as required for cell migration [49]. This includes actin remodelling to form filopodia and lamellipodia, stress fibre formation, and focal adhesion turnover [41]. A crucial consequence of such remodelling is that ECs acquire front-to-rear polarity which coincides with the direction of their migration [50]. Membrane protrusions, such as filopodia and lamellipodia, increase the cell surface area at the leading edge. More VEGF receptors and integrins, which bind to ECM components, become activated at the cell front to further reinforce the cytoskeleton remodelling and polarisation [51]. This mechanism is known as taxis (e.g. chemotaxis, haptotaxis) and allows ECs to sense extracellular cues and migrate towards them. The cells' chemotactic sensitivity leads to the so-called *brush-border* effect which is

characterised by increased rates of branching and higher EC densities closer to the source of the VEGF stimulus which can be sensed by cells and biases their migration towards higher VEGF concentrations.

Whilst interactions with ECM components play a central role in EC polarisation and movement, EC migration, in turn, reorganises and remodels the ECM [45]. Prior to assembly of the basement membrane of the newly formed vessels, the ECM microenvironment consists mostly of collagen I and elastin fibers. Activated ECs secrete matrix type 1 metalloproteinases (MT1-MMPs) that degrade the ECM [44, 46, 52]. This process generates ECM-free tunnels into which the sprouts can elongate [52]. As sprouts grow, they assemble a basement membrane which contains, among other things, fibrous components (collagen IV, fibronectin and various laminins [44]) that are secreted by the ECs and whose function is to promote cell-cell and cell-ECM contact and to limit EC migration [46].

Further matrix reorganisation occurs in response to the mechanical forces generated by migrating cells as they realign the collagen fibrils of the ECM. Several experiments have shown that cells with extended filopodia and lamellipodia form focal adhesions with the ECM components and realign them by pulling in the direction parallel to their motion [39, 40, 51, 53]. They also move the fibrils from the local neighbourhood closer to their surfaces. As a result, collagen fibrils accumulate and align along the direction of sprout elongation. Since cell-followers form focal adhesions with these aligned fibrils they automatically polarize and migrate in the direction of sprout elongation [42, 46, 47]. In this way sprout integrity is maintained and the coordinated motion of ECs emerges.

The default phenotype of an EC that has been activated by VEGF is a "tip" cell [12, 13]. Tip cells are characterised by exploratory behaviour and low proliferative activity. Thus, although the ECM regulates EC behaviour to a great extent [45], if all cells were to move at the same time in an exploratory fashion, sprout integrity would be lost. In non-pathological angiogenesis, typical vascular network morphology is obtained when EC proliferation and migration (i.e. the exploration of space by active ECs in response to signalling cues) are properly balanced. This balance is mediated by the Delta-Notch signalling pathway, which provides ECs with a contact-dependent cross-talk mechanism by which they can acquire a "stalk" cell phenotype [12, 13]; stalk cells are characterised by reduced migratory and higher proliferative activities. In this way, cell-cell communication allows cells to adjust their gene expression patterns (and, hence, phenotype) and to coordinate their behaviour [54]. The processes influenced by cell phenotype in our model are illustrated in Fig 1A. The ratio between the number of tip and stalk cells during sprouting plays a key role in the integrity of the developing vascular network and its functionality. This was confirmed by experiments in which Notch activity was inhibited or completely blocked. This caused all cells to adopt a tip cell phenotype and led to pathological network formation [7].

## Mathematical modelling of random cell migration

Stochastic approaches have been successfully used to mathematically model systems with random, heterogeneous behaviour, including spatially extended systems [27, 55–64]. We model cell migration at the cellular scale stochastically to account for random cell motility, cell mixing, and branching dynamics. This approach allows us to compare our model with *in vitro* experiments which, as any biological system, are noisy.

Within the context of spatially extended systems, a stochastic model can be formulated as an individual-based, or a compartment-based, model. In the former approach, the Brownian dynamics of each individual agent (cells, molecules, etc.) are simulated, which becomes computationally expensive as the number of agents increases. The latter is more numerically

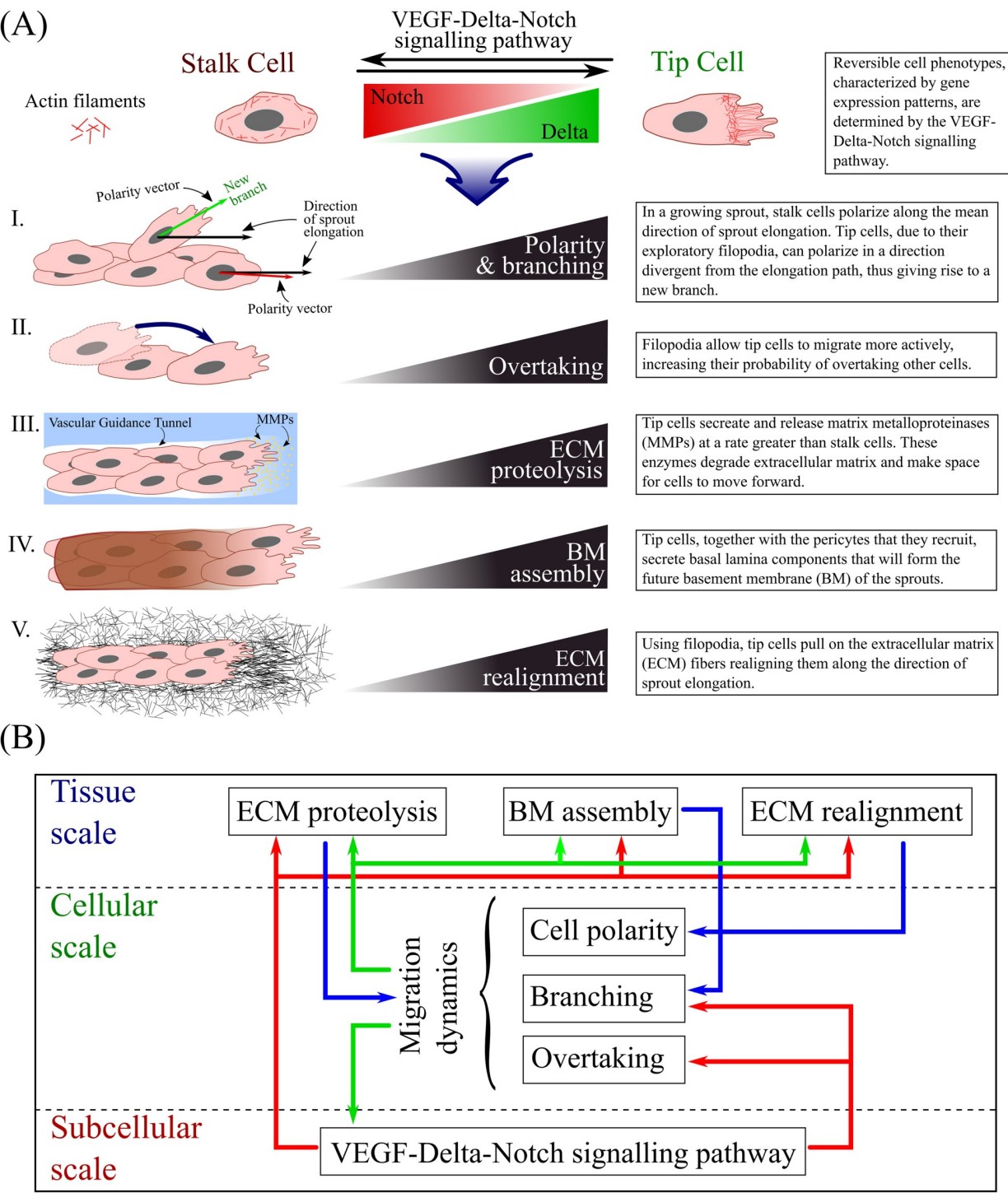

**Fig 1. Schematic summary of our model.** (A) The gene expression profile of each cell (and, hence, its phenotype) is accounted for at the subcellular scale of our model. It influences such processes as I. cell polarity and branching, II. overtaking, III. extracellular matrix (ECM) proteolysis, IV. basement membrane (BM) assembly and V. ECM realignment. The cartoons on the left illustrate each process, whereas the text-boxes on the right provide brief descriptions of how they are influenced by tip and stalk cells. (B) The structure of our multiscale model. The diagram illustrates the processes that act at each spatial scale. Arrows illustrate coupling between the scales.

efficient for the purpose of simulating small vascular networks containing at most hundreds of cells. In this method the domain is partitioned into non-overlapping compartments, or voxels so that the position of each agent is known up to the voxel size. Agents within the same voxel are considered indistinguishable and reactions between them occur independently of those in other compartments. Movement of agents between the voxels is modelled as a continuous time random walk (RW). This is also known as the Reaction Diffusion Master Equation (RDME) approach.

A weakness of the RDME approach is that it does not converge to its individual-based Brownian dynamics in dimensions greater than one for compartments with size less than a specific lower bound (of the order of the reaction radius in the Smoluchowski interpretation) [65]. The RDME breaks down as the voxel size tends to zero because the waiting time for multi-molecular reactions becomes infinite. One strategy to address this problem, the so-called convergent RDME (cRDME), is an approximation of Doi's model for the binary reactions of the form $A + B \rightarrow C$ proposed in [66]. In the cRDME, agents from *different* voxels may interact via multi-molecular reactions provided they lie within a predefined *interaction radius*. In this way the artefact of vanishing reaction rates as voxel size tends to zero is avoided.

We formulate the cellular scale of our model using the cRDME approach. We introduce an interaction radius, $R_c$, within which cells are assumed to interact. This approach also fits naturally with the VEGF-Delta-Notch driven cell cross-talk that we incorporate at the subcellular scale of our multi-scale model (see below).

For the numerical implementation of the cRDME we use the Next Subvolume (NSV) method [67], which is a computationally efficient implementation of the standard Stochastic Simulation Algorithm [68] to simulate RDME/cRDME.

## Summary of the multiscale model

The model we develop is a two-dimensional stochastic multiscale model of migration-driven sprouting angiogenesis. Its structure is shown in Fig 1B. Briefly, the model operates on three distinct spatial scales:

- The **subcellular scale** defines the gene expression pattern of individual cells. Since ECs contain a finite number of proteins, some level of noise is always present in the system. Thus, we implement a stochastic model of the VEGF-Delta-Notch signalling pathway to describe the temporal dynamics of the number of ligands/receptors for each cell. This pathway is known to produce bistable behaviour. Cells exhibit either high Delta and VEGFR2, and low Notch levels (the tip phenotype) or low Delta and VEGFR2, and high Notch levels (the stalk phenotype). Stochasticity allows random transitions between these phenotypes in regions of bistability, behaviour which cannot be achieved in a deterministic model.

- The **cellular scale** accounts for cell migration. It is formulated as a variant of an on-lattice persistent random walk (PRW) of ECs.

- The **tissue scale** keeps track of the local ECM environment of the cells. Local ODEs track the evolution of the concentrations of the existing ECM and BM components, whereas ECM fibril alignment driven by EC movement is updated using a phenomenological model.

An illustration of the model geometry can be found in Fig 2.

In general, an EC has an arbitrary shape which depends on its cell-cell and cell-matrix focal adhesions. In our model we do not keep track of the exact cell shape and assume that cell position is known up to the position of its nucleus. Thus, when referring to a cell position, we refer to the position of its nucleus and assume that the cell has some arbitrary shape centred on the

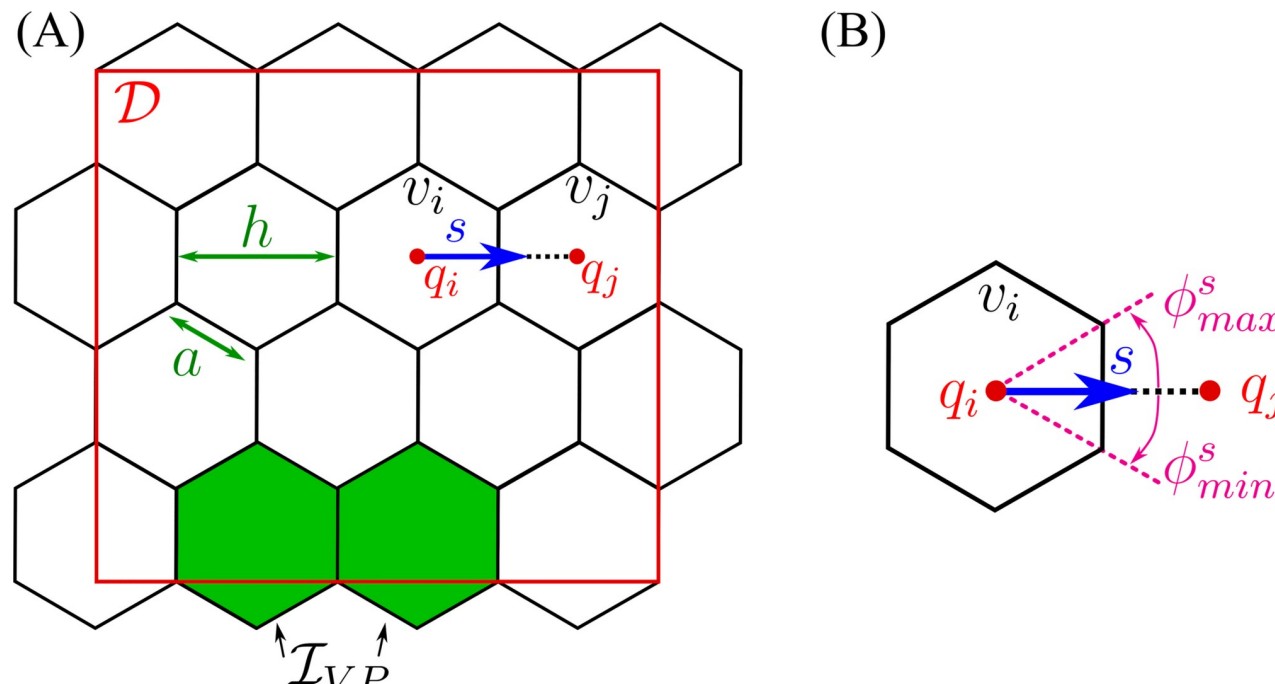

**Fig 2. Model geometry. (A)** To account for cell migration in the framework of a PRW, we cover the domain, $\mathcal{D} \subset \mathbb{R}^2$, by a uniform lattice $\mathcal{L} = \{v_k \; : \; \bigcup_{k=1}^{N_I} v_k \supset \mathcal{D}\}$ of non-overlapping hexagonal voxels, $v_k$, of width $h$ (or hexagon edge, $a$, $h = \sqrt{3}a$). We denote by $\mathcal{I} = \{k = (k_x, k_y)^T \; : \; v_k \in \mathcal{L}\}$ the set of 2D indices, with cardinal, $|\mathcal{I}| = N_I$. The coordinates of the centre of voxel $v_k$ are denoted by $q_k = (q_k^x, q_k^y)^T$. We assume that there is a constant supply of ECs to the domain, coming from the vascular plexus. These ECs enter the domain at fixed boundary voxels, defined as a set, $\mathcal{I}_{VP}$ (coloured in green). **(B)** A detailed illustration of an individual voxel, $v_i$. $s = h^{-1}(q_j - q_i)$ denotes a normalised vector of the migration direction between neighbouring voxels, $v_i$ and $v_j$. There are at most 6 possible migration directions for each hexagonal voxel. Each migration direction, $s$, can be characterised by an equivalent angle interval $[\phi_{min}^s, \phi_{max}^s]$.

voxel containing its nucleus (see Fig 3A). A consequence of this approach is that, since cells can extend membrane protrusions and interact with distinct cells beyond their first neighbours, interactions between cells in our model are non-local. We introduce two interaction radii, $R_s$ and $R_c$, for the subcellular and cellular scales, respectively, and assume that a cell can interact with any neighbouring cell partially overlapped by a circular neighbourhood of these interaction radii (see Fig 3A). In particular, at the subcellular scale trans-binding between a Delta ligand on one cell and a Notch receptor on another can occur if the distance between their cell centres is less than the interaction radius, $R_s$. Thus, the total amount of ligand/receptor (belonging to a neighbour/neighbours) to which a cell is exposed is proportional to the surface area of the overlap region between the circular neighbourhood and the neighbouring voxel/voxels (not necessarily first neighbours). A similar modelling technique is employed at the cellular scale with the interaction radius, $R_c$, which we use to account for cell-cell adhesion. These radii are not necessarily equal (although they are of the same order of magnitude) since different types of interactions are considered at each scale.

We perform our numerical simulations over time periods that are commensurate with the duration of *in vitro* experiments (hours) and for which proliferation is negligible [2, 7, 54]. For these reasons the model focuses only on the coordinated migration of ECs and assumes that proliferation occurs only at the vascular plexus [54], the initial vascular bed. This effect is implemented by introducing ECs into the domain at a specific set of boundary voxels, $\mathcal{I}_{VP}$ (see Fig 2A).

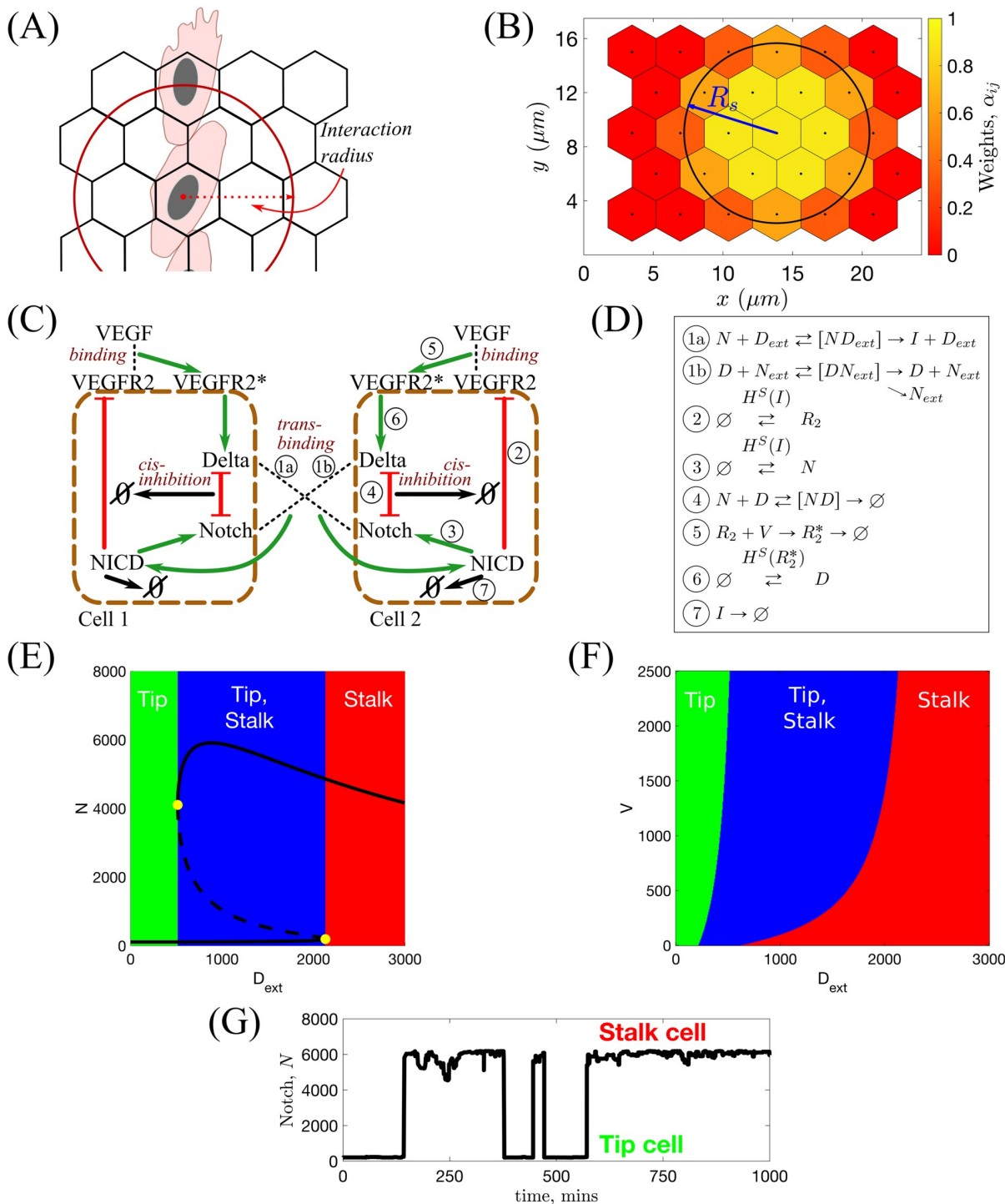

**Fig 3. Illustrations for the subcellular scale of the model.** (A) Since we track the position of cells only up to the positions of their nuclei, interactions are assumed to be non-local within some *interaction radius*. The subcellular and cellular scale interaction radii are denoted by $R_s$ and $R_c$, respectively. (B) Illustration of values of the weight coefficients, $\alpha_{ij}$ (see Eq (5)). (C) Schematic illustration of the VEGF-Delta-Notch signalling pathway. NICD stands for Notch intracellular domain, VEGFR2 for VEGF Receptor 2 and VEGFR2* for activated, i.e. bound to VEGF, VEGFR2. In this case, the local environment of Cell 2 is Cell 1, and $D_{ext}$ and $N_{ext}$ correspond, respectively, to the Delta and Notch concentrations in Cell 1 (and vice versa). Circled numbers correspond to the kinetic reactions listed in (D). (D) Kinetic reactions used for the VEGF-Delta-Notch pathway. See Table 1 for description of the model variables. $H^S(var)$ indicates that the transition rate of gene expression of a protein is transcriptionally regulated by the signalling variable, *var*. Here, $H^S(\cdot)$ is the shifted Hill function (see caption of Table 2). Simple arrows indicate reactions with constant rates. (E) Bifurcation diagram of Notch concentration, $N$, as a function of external Delta ligand, $D_{ext}$,

corresponding to the system of equations Eq (1). Full lines denote stable steady states; dashed lines—unstable steady state; yellow filled dots—saddle-node bifurcation points. **(F)** Phenotype diagram as a function of external Delta, $D_{ext}$, and external VEGF, $V$, corresponding to the system of equations Eq (1). **(G)** In simulations, the local environment of a cell (Delta and Notch levels in a neighbourhood of the cell, $D_{ext}$ and $N_{ext}$ (Eq (6))) dynamically changes with time due to cell migration. This leads to phenotype switches.

Since the experimental data we use for model calibration and validation were extracted from experiments carried out with constant VEGF concentration supplied externally, we assume that the distribution of VEGF is maintained at a (prescribed) constant value at all times.

Interested readers will find more detail on model formulation later in this section and in S1 Text.

We list the variables of our multi-scale model in Table 1.

**Table 1. The description of the model variables.** The variables in the table are organised by spatial scales. Subcellular variables, used to simulate the VEGF-Delta-Notch signalling pathway, define the gene expression pattern of individual cells (their phenotypes). Cellular scale variables define the occupancy of the lattice by ECs. Tissue scale variables define the composition and structure of the ECM. Bold letters denote vector variables specifying variable configuration for the whole lattice; normal font letters correspond to the variables associated with a particular voxel, the index of which is specified by a subscript.

| | Variable | Description |
|---|---|---|
| **Subcellular scale** | $\boldsymbol{N} = (N_1, \ldots, N_{N_I})$ | The distribution of Notch receptor among voxels. If there is no cell nucleus in a voxel $v_i$, i.e. $E_i = 0$, then $N_i = 0$. |
| | $\boldsymbol{D} = (D_1, \ldots, D_{N_I})$ | The distribution of Delta ligand among voxels. If there is no cell nucleus in a voxel $v_i$, i.e. $E_i = 0$, then $D_i = 0$. |
| | $\boldsymbol{I} = (I_1, \ldots, I_{N_I})$ | The distribution of Notch intracellular domain (NICD) among voxels. If there is no cell nucleus in a voxel $v_i$, i.e. $E_i = 0$, then $I_i = 0$. |
| | $\boldsymbol{R2} = (R2_1, \ldots, R2_{N_I})$ | The distribution of VEGFR2 receptor among voxels. If there is no cell nucleus in a voxel $v_i$, i.e. $E_i = 0$, then $R2_i = 0$. |
| | $\boldsymbol{R2}^* = (R2_1^*, \ldots, R2_{N_I}^*)$ | The distribution of activated VEGFR2 receptor (VEGFR2 bound to extracellular VEGF) among voxels. If there is no cell nucleus in a voxel $v_i$, i.e. $E_i = 0$, then $R2_i^* = 0$. |
| **Cellular scale** | $\boldsymbol{E} = (E_1, \ldots, E_{N_I})$ | The distribution of EC nuclei among voxels. $E_i = 1$ if a cell nucleus is present in the voxel $v_i$, $E_i = 0$, otherwise. At most one cell nucleus is allowed per voxel. |
| | $\boldsymbol{E}^N = (E_1^N, \ldots, E_{N_I}^N)$ | The neighbourhood nucleus distribution. This variable is completely defined by the configuration of $\boldsymbol{E}$. Each $E_i^N = \dfrac{\sum_{j \neq i} \phi_{ij} E_j}{\sum_{j \neq i} \phi_{ij}}$, where $\phi_{ij} = \dfrac{|v_j \cap \mathcal{B}_{R_c}(i)|}{|v_j|}$, $\mathcal{B}_{R_c}(i)$ is a circular neighbourhood of interaction radius, $R_c$, centred in the voxel $v_i$, and $|\cdot|$ denotes area. |
| **Tissue scale** | $\boldsymbol{c} = (c_1, \ldots, c_{N_I})$ | The ECM density, consisting mostly of collagen I and elastin fibers. It is degraded by cells via a process termed *ECM proteolysis* (Fig 1A III.). We assume $0 \leq c_i \leq c_{max}$ for all $i \in \mathcal{I}$, where $c_{max} > 0$ is a parameter characterising the maximum density of the ECM. |
| | $\boldsymbol{m} = (m_1, \ldots, m_{N_I})$ | The concentration of basal lamina components (collagen IV, fibronectin and various laminins) newly deposited by cells that are used for *BM assembly* (Fig 1A IV.). We assume $0 \leq m_i \leq 1$ for all $i \in \mathcal{I}$; $m_i = 0$ if no BM components have been deposited yet, whereas $m_i = 1$ if a BM has been assembled around the sprout segment situated in voxel $v_i$. |
| | $\boldsymbol{l} = (l_1, \ldots, l_{N_I})$ | The orientation landscape (OL) variable representing the *alignment of ECM fibrils* (Fig 1A V.) within the voxel. For a hexagonal lattice $l_i = \{l_i^s\}_{s \in \mathcal{S}}$ for all possible jumping directions $s \in \mathcal{S} = \{r, ur, ul, l, dl, dr\}$ (*r*—right, *ur*—upward-right, *ul*—upward-left, *l*—left, *dl*—downward-left, *dr*—downward-right), $s \in \mathbb{R}^2$ (see Fig 2). An example of possible orientation landscape configurations is shown in Fig 5A. |

**Subcellular scale. VEGF-Delta-Notch model of phenotype selection.**   At the subcellular scale we account for the VEGF-Delta-Notch signalling pathway which determines the gene expression pattern (phenotype) of each EC (see Fig 3C). This pathway mediates inter-cellular cross-talk and typically produces alternating patterns of tip and stalk cell phenotypes within growing sprouts [11]. In our model, the pathway is simulated via a *bistable* stochastic system which accounts for intrinsic noise and, in particular, noise-induced random transitions between the stable steady states of the system (phenotypes) [69–72]. We posit that such phenotypic switches are essential for understanding the complex dynamics of ECs within growing sprouts [2]. Our subcellular model is based on previous work [73–76]. Following [75, 76], we combine the lateral inhibition model of the Delta-Notch signalling pathway introduced in [73, 74] with the VEGF signalling pathway. The Delta-Notch model accounts for cis-inhibition when a Delta ligand and Notch receptor from the same cell inhibit each others' activity. We include this interaction since cis-inhibition has been shown to substantially speed up phenotype specification [77].

**Individual cell system.**   The kinetic reactions acting on individual cells system are illustrated in Fig 3D. We account for *trans*-activation of Notch receptor (production of an NICD) when it *trans*-binds to a Delta ligand belonging to a neighbouring cell, $D_{ext}$ (reaction (1a)). If a Delta ligand *trans*-binds to a Notch receptor on a neighbouring cell, $N_{ext}$, it is either endocytotically recycled or degraded (reaction (1b)). In this reaction, we assume that the active Notch signal is produced in the neighbouring cell, the dynamics of which are irrelevant for the cell of interest. Once cleaved from the Notch receptor, active Notch signal, NICD, is translocated to the cell nucleus where it down-regulates gene expression of VEGFR2 (reaction ②) and up-regulates gene expression of the Notch receptor (reaction ③). *Cis*-inhibition is accounted for in reaction ④ in which mutual inhibition is assumed for a Delta ligand and a Notch receptor interacting within the same cell. External VEGF, $V$, can bind to and activate a VEGFR2 (reaction ⑤). This leads to up-regulation of Delta production (reaction ⑥). Reaction ⑦ corresponds to degradation of NICD.

An essential feature of our subcellular model is that it exhibits bistability. To demonstrate this, we derived the mean-field limit equations associated with the kinetic reactions shown in Fig 3D (see Eq (1) in Table 2) and performed a numerical bifurcation analysis (see S1 Table for a list of the parameter values). The results presented in Fig 3E show how the steady state value of the Notch concentration, $N$, changes as the concentration of external Delta ligand, $D_{ext}$, varies. For small (large) values of $D_{ext}$, the system supports a unique steady-state corresponding to the tip (stalk) phenotype. For intermediate values of $D_{ext}$, the system is bistable: both phenotypes coexist. The combined effect of the external VEGF, $V$, and $D_{ext}$ on the system is shown in Fig 3F. We see that varying $V$ does not alter the qualitative behaviour shown in Fig 3E, although the size of the bistable region decreases as $V$ decreases.

The stable states of the subcellular system are characterised by distinct gene expression patterns (for example, high (low) Delta level, $D$, corresponds to tip (stalk) phenotype). Thus, one variable suffices in order to effectively define the phenotypes. We use Delta level, $D$, as a proxy variable in the following way

$$\text{tip phenotype}: \quad D \geq b_D,$$
$$\text{stalk phenotype}: \quad \text{otherwise}, \tag{3}$$

where $b_D$ is baseline gene expression of Delta ligand in ECs (see S1 Table).

**Multicellular system.**   In order to account for cell-cell cross-talk via the VEGF-Delta-Notch pathway, we extend the individual cell system (see Fig 3D) to a multicellular environment

**Table 2. Mean-field equations associated with the stochastic system of the VEGF-Delta-Notch signalling pathway for the individual cell system (left column) and the multicellular system (right column).** Here $H^S(\cdot)$ is the so-called shifted Hill function [78]. Its functional form is given by $H^S(X) = H^S(X; X_0, \lambda_{X,Y}, n_Y) = \frac{1 + \lambda_{X,Y}(X/X_0)^{n_Y}}{1 + (X/X_0)^{n_Y}}$, where $X_0$, $\lambda_{X,Y}$ and $n_Y$ are positive parameters (see S1 Text for more details). Description and values of parameters can be found in S1 Table.

| Individual cell system (1) | Multicellular system (2) |
|---|---|
| $\dfrac{dN}{dt} = b_N H^S(I; I_0, \lambda_{I,N}, n_N) - \gamma N - k_t D_{ext} N - k_c ND,$ | $\dfrac{dN_i}{dt} = b_N H^S(I_i; I_0, \lambda_{I,N}, n_N) - \gamma N_i - k_t N_i \bar{D}_i - k_c N_i D_i,$ |
| $\dfrac{dD}{dt} = b_D H^S(R2^*; R2_0^*, \lambda_{R2^*,D}, n_D) - \gamma D - \eta k_t N_{ext} D - k_c ND,$ | $\dfrac{dD_i}{dt} = b_D H^S(R2_i^*; R2_0^*, \lambda_{R2^*,D}, n_D) - \gamma D_i - \eta k_t D_i \bar{N}_i - k_c N_i D_i,$ |
| $\dfrac{dI}{dt} = k_t D_{ext} N - \gamma_e I,$ | $\dfrac{dI_i}{dt} = k_t N_i \bar{D}_i - \gamma_e I_i,$ |
| $\dfrac{dR2}{dt} = b_{R2} H^S(I; I_0, \lambda_{I,R2}, n_{R2}) - \gamma R2 - k_v VR2,$ | $\dfrac{dR2_i}{dt} = b_{R2} H^S(I_i; I_0, \lambda_{I,R2}, n_{R2}) - \gamma R2_i - k_v R2_i V,$ |
| $\dfrac{dR2^*}{dt} = k_v VR2 - \gamma_e R2^*,$ | $\dfrac{dR2_i^*}{dt} = k_v R2_i V - \gamma_e R2_i^*,$ |
| $D_{ext}, N_{ext}$ are constant input parameters. | $D_{ext} = \bar{D}_i,\ N_{ext} = \bar{N}_i$ are given by (6), $i \in \mathcal{I}.$ |

by specifying for each cell the external (i.e. belonging to neighbouring cells) amount of Delta and Notch to which it is exposed. As mentioned above, since cell positions in our model are only known up to the position of their nuclei, we assume non-local interactions between cells within a reaction radius, $R_s$. Thus, we define the local environment of a cell whose nucleus is situated in voxel $v_i$ as the set of voxels with a non-zero overlap region with a circular neighbourhood of radius $R_s$ centred at voxel $v_i$, $\mathcal{B}_{R_s}(i)$,

$$H(i) := \{v_j : v_j \cap \mathcal{B}_{R_s}(i) \neq \emptyset, \ j \neq i, \ j \in \mathcal{I}\}. \tag{4}$$

The weights, $\alpha_{ij}$, assigned to each voxel $v_j \in H(i)$, (see Fig 3B) are defined as follows

$$\alpha_{ij} = \frac{|v_j \cap \mathcal{B}_{R_s}(i)|}{|v_j|}, \quad i, \ j \in \mathcal{I}, \tag{5}$$

where $|\cdot|$ denotes 2D area.

The external Delta (Notch) concentration, $D_{ext}$ ($N_{ext}$), for a cell situated in a voxel $v_i$ is defined as follows:

$$
\begin{aligned}
D_{ext} &= \overline{D_i} = \frac{\displaystyle\sum_{v_j \ \in \ H(i)} \alpha_{ij} D_j}{\displaystyle\sum_{v_j \ \in \ H(i)} \alpha_{ij}}, \\[2em]
N_{ext} &= \overline{N_i} = \frac{\displaystyle\sum_{v_j \ \in \ H(i)} \alpha_{ij} N_j}{\displaystyle\sum_{v_j \ \in \ H(i)} \alpha_{ij}},
\end{aligned}
\tag{6}
$$

where $D_j$ ($N_j$) denotes the Delta (Notch) concentration in voxel $v_j$.

Therefore, our multicellular stochastic system at the subcellular scale consists of the same kinetic reactions as in Fig 3D formulated for each voxel $v_i$, $i \in \mathcal{I}$ with $D_{ext}$, $N_{ext}$ given by (6). It is important to note that, in simulations of angiogenic sprouting, the quantities $D_{ext}$ and $N_{ext}$ dynamically change due to cell migration, thus leading to phenotype re-establishment (see Fig 3G).

When simulated within a two-dimensional domain, our multicellular system produces an alternating pattern of tip/stalk phenotypes. As the reaction radius, $R_s$, changes the system dynamics do not change but the proportion of tip and stalk cells does. To illustrate this, we ran simulations of the stochastic multicellular system for a $10 \times 12$ regular monolayer of cells for different values of the interaction radius, $R_s$ (see S1 Fig). These simulations revealed that the distance between tip cells increases (i.e. the proportion of tip cells decreases) as $R_s$ increases. We also investigated how phenotype patterning changes within a monolayer as cis-inhibition intensity varies (see S1 Text). For low values of the cis-inhibition parameter, $\kappa_c$, typical chessboard tip-stalk pattern is produced; as $\kappa_c$ increases, ECs with tip phenotype can become adjacent to each other, thus increasing the time to patterning, since the lateral inhibition is weakened (see S1 Text).

The mean-field equations associated with the multicellular stochastic system are given by Eq (2) in Table 2.

More detail on the derivation and analysis of the subcellular model can be found in S1 Text.

**Cellular scale. Persistent random walk and cell overtaking.**   At the cellular scale we account for EC migration and overtaking. In more detail, we consider a compartment-based

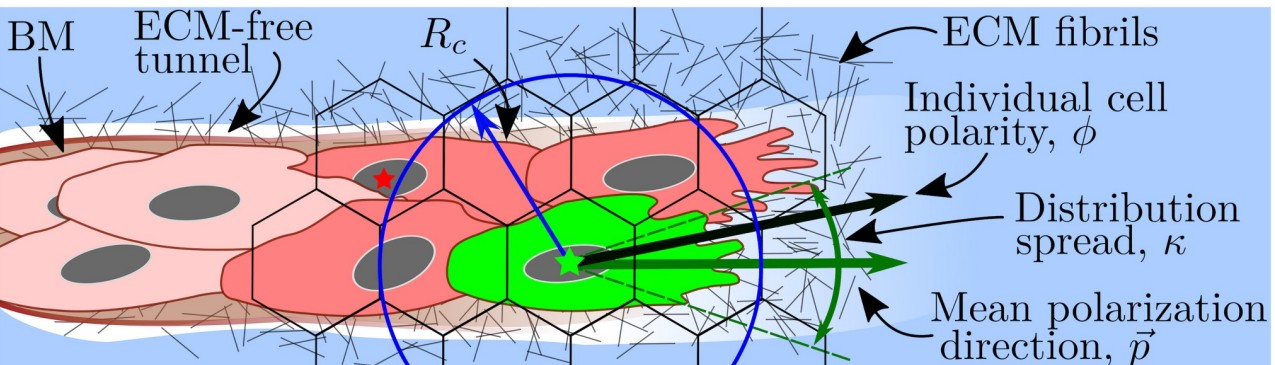

**Fig 4. A cartoon illustrating cell migration.** Migration transition rate (Eq (7)) is illustrated for a cell coloured in green. Its nucleus is marked with a green star. The motility of this cell depends on the ECM density, $c_i$, in the voxel where its nucleus is situated (accounted for by the $S(\cdot)$ function (Eq (8)). The green cell forms cell-cell adhesions with other cells coloured in red. In this work, we assume a circular neighbourhood for cell-cell interactions within the so-called *interaction radius*, $R_c$, for the cellular scale (here it is drawn to have the same value as in our simulations, $R_c = 1.5h$, where $h$ is the voxel width). This allows cells to interact beyond their immediate neighbours as, for example, the cell with the nucleus marked by a red star and the focal green cell. Other geometries (e.g. elliptical neighbourhood aligned with cell polarity vector) are unlikely to alter the model behaviour significantly since, in sprouting structures, lateral regions of the circular neighbourhood, which would be ignored by an elliptic neighbourhood, are typically empty. Cell-cell adhesion is accounted for by the *neighbourhood function*, $F(\cdot)$, Eq (9). The *individual cell polarity*, $\phi$, is sampled from the von Mises distribution (Eq (12)) with the mean value given by the *mean polarization direction*, $\vec{p}$ (calculated as a function of local ECM fibril alignment, $l_i$, Eq (13)). The *distribution spread*, $\kappa$, is assumed to depend on the focal cell phenotype and the concentration of the BM components, Eq (14).

model of a persistent random walk (PRW) [79], in which transition rates depend on the phenotypic state of individual cells and their interactions with ECM components.

We denote by $s = h^{-1}(q_j - q_i) \in \mathcal{S}$ the migration direction, as a unit vector pointing from $q_i$ towards $q_j$, the centres of neighbouring voxels $v_i$ and $v_j$, respectively (see Fig 2B). In order to formulate the PRW of ECs, we introduce, $\omega(i \to j)$, to denote the probability that a cell moves from voxel $v_i$ to a neighbouring voxel $v_j$ (i.e. along the direction $s$) where

$$\omega(i \to j) = \overbrace{\widetilde{D_\omega}}^{\substack{\text{cell} \\ \text{occupancy}}} E_i \overbrace{S(c_i)}^{} \overbrace{F\left(E_i^N\right)}^{\substack{\text{cell-cell} \\ \text{adhesion}}} \rho_{ij}(D_i) \overbrace{\left[\int_0^{2\pi} W^s(\phi) f_{vM}(\phi|\theta,\kappa)\,d\phi\right]}^{\text{cell polarity}}. \tag{7}$$

$$\underbrace{\phantom{\widetilde{D_\omega}}}_{\substack{\text{diffusion} \\ \text{coefficient}}} \underbrace{\phantom{S(c_i)}}_{\substack{\text{ECM} \\ \text{density}}} \underbrace{\phantom{\rho_{ij}(D_i)}}_{\substack{\text{overtaking} \\ \text{probability}}}$$

We explain below each of the terms that appears in Eq (7) (see also Fig 4 for a graphical illustration and Table 1 for a description of the model variables).

**Diffusion coefficient.** Transition rates in the framework of a PRW must be appropriately scaled with the size of a voxel (depending on the particular lattice used to discretise the domain). For a uniform hexagonal lattice, $\mathcal{L}$, the diffusion coefficient of the transition $\omega(i \to j)$ is scaled as $\widetilde{D_\omega} = {D_\omega}/{h^2}$ [80]. Here $h[\mu m]$ is the width of a hexagonal voxel (see Fig 2) and $D_\omega \left[\frac{\mu m^2}{min}\right]$ is the macroscale diffusion coefficient for the ECs.

**Cell occupancy.** If a cell nucleus is present in voxel $v_i$, then $E_i = 1$ (at most one cell nucleus is allowed per voxel). If the voxel $v_i$ is empty, then $E_i = 0$ and $\omega(i \to j) = 0$.

**ECM density.** The function $S(c_i)$ accounts for the effect of the local ECM density, $c_i$, on cell motility. In general, $S(\cdot)$ is a decreasing function of its argument. We assume further that ECs cannot move if the ECM concentration exceeds a threshold value, $c_{max}$ ($S(c_i) = 0$ for

$c_i \geq c_{max}$). In our simulations we fix

$$S(c_i) = \begin{cases} 1 - \dfrac{c_i}{c_{max}} & \text{if } 0 \leq c_i < c_{max} \\ 0, & \text{otherwise.} \end{cases} \qquad (8)$$

**Cell-cell adhesion.** The effect of cell-cell adhesion on cell migration is incorporated via the so-called *neighbourhood function*, $F(E_i^N)$. Its argument, $E_i^N$, represents the number of ECs in a cell's local neighbourhood (red-coloured cells in Fig 4). The functional form of the neighbourhood function was chosen in order to phenomenologically capture the way in which EC behaviour depends on the cell-cell contacts (see Fig 5E). In biological experiments, it was shown that when a cell loses contact with its neighbours (laser ablation experiments in [8]) it halts until the following cells reach it. This is captured by the increasing part of $F(E_i^N)$; when the number of neighbours around a cell is below the first threshold, $E_{F1}$, the probability of cell movement decreases rapidly to zero, thus the migration transition (Eq (7)) goes to 0 as well. Similarly, when there are many ECs in a cell's neighbourhood, its movement slows down. This is accounted for by the decreasing part of $F(E_i^N)$ when the number of neighbouring cells exceeds the second threshold value, $E_{F2}$, cell movement is slowed down and eventually halts in regions of high cell density.

$$F(E_i^N) = \left( \frac{1}{1 + \exp\left(-s_{F1}(E_i^N - E_{F1})\right)} + \frac{1}{1 + \exp\left(s_{F2}(E_i^N - E_{F2})\right)} - 1 \right)^+, \qquad (9)$$

where $(x)^+ = max(0, x)$, and the parameters $E_{F1}$, $E_{F2}$, $s_{F1}$ and $s_{F2}$ characterise the shape of the curve.

**Overtaking probability.** This term accounts for cell overtaking and excluded volumes. A jump occurs with overtaking probability, $\rho_{ij}(D_i) = 1$, if the target voxel, $v_j$, is empty ($E_j = 0$). Otherwise, the cells in voxels $v_i$ and $v_j$ switch their positions with probability $\rho_{ij}(D_i) = p_{switch}(D_i)$. We consider this probability to be phenotype dependent. In particular, we assume that tip cells (see Eq (3)) are more motile because their filopodia are stronger (see Fig 1A II.). Thus, the switching probability, $p_{switch}(D_i)$, is assumed to be an increasing function of the Delta level, $D_i$, of the migrating cell

$$\rho_{ij}(D_i) = \begin{cases} 1, & \text{if } E_j = 0, \\ p_{switch}(D_i), & \text{otherwise.} \end{cases} \qquad (10)$$

where

$$p_{switch}(D_i) = \frac{p_{max}}{1 + \exp\left(-s_p(D_i - D_p)\right)}, \qquad (11)$$

the parameters $s_p$ and $D_p$ characterise, respectively, the slope and position of the sigmoid, and $p_{max}$ denotes its maximum value (see Fig 5F).

**Cell polarity.** Prior to migration, cells develop a polarity which depends on their local environment. Following [81, 82], we consider the local cell polarity, $\phi$, (see Fig 4) to be a random quantity sampled from the von Mises distribution. The probability density function (pdf) of the von Mises distribution reads

$$f_{vM}(\phi | \mu, \kappa) = \frac{\exp\left(\kappa \cos\left(\phi - \mu\right)\right)}{2\pi I_0(\kappa)}. \qquad (12)$$

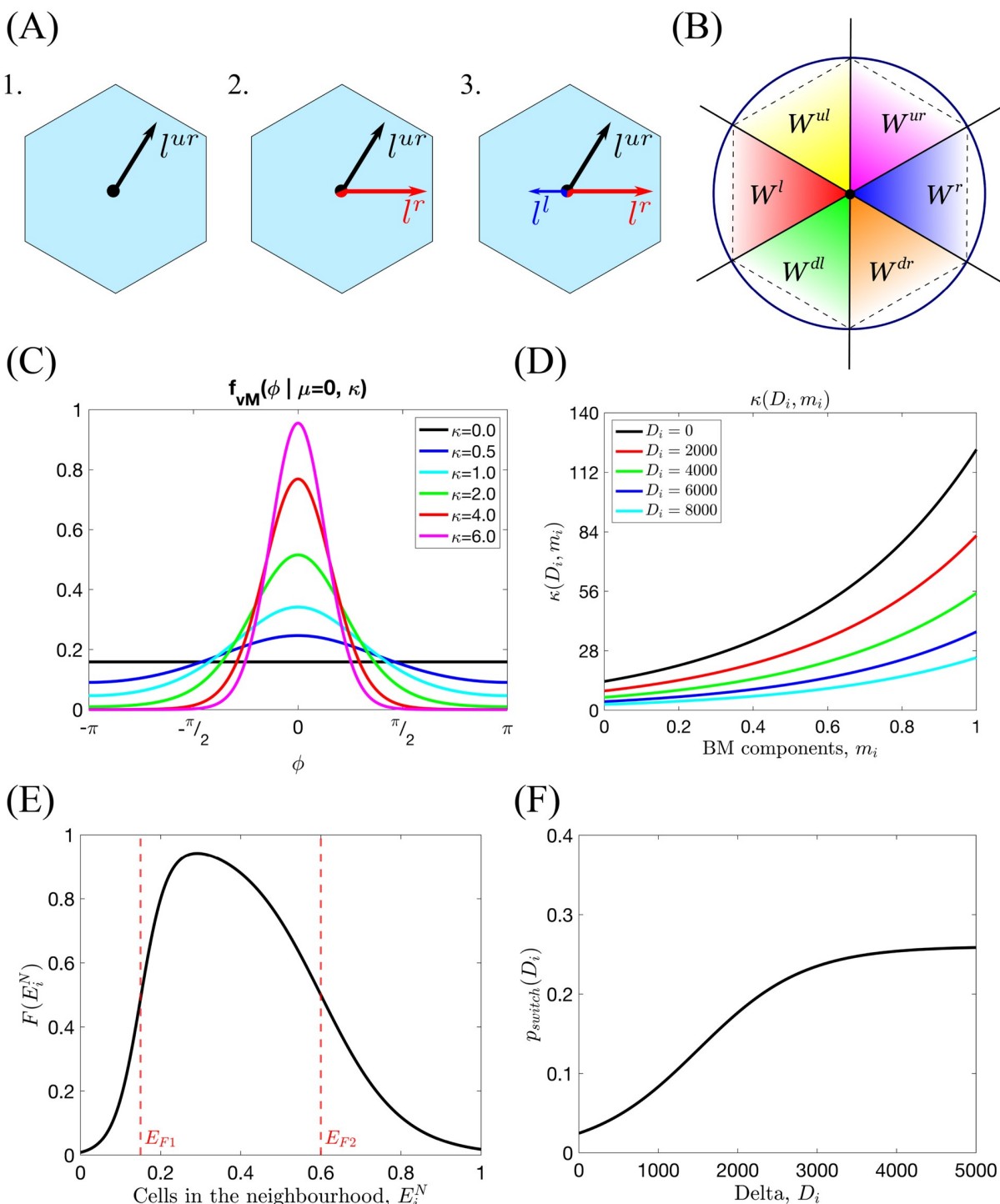

**Fig 5. Series of sketches illustrating the components of the cellular scale persistent random walk. (A)** An example of the orientation landscape, $l$, configurations for a hexagonal lattice. **1.** All fibrils are aligned in the upward-right direction; this would be an example of a strongly aligned part of the ECM. **2.** Half of the fibrils are aligned to the right, half in the upward-right direction. This would correspond to a branching point. **3.** As in **2.** but with some additional fibrils aligned in the left direction. **(B)** The window function, $W^s(\phi)$, (Eq (15)) has been defined as an indicator function over an angle interval corresponding to each possible migration direction $s \in \mathcal{S}$ (lattice-dependent). The diagram illustrates these intervals for a hexagonal lattice. **(C)** Illustration of the probability distribution function of the von Mises distribution, $f_{vM}(x|\mu, \kappa)$, centred at $\mu = 0$ for different fixed $\kappa$ (Eq (12)). **(D)** Illustration of the $\kappa$ function as a function of local Delta ligand level, $D_i$, and concentration of BM components, $m_i$, (Eq (14) with $K = 13.6$, $k_m = 2.2$ and $k_D = 0.0002$). **(E)** An example of the neighbourhood function, $F(E_i^N)$ (Eq (9)), with $E_{F1} = 0.15$, $E_{F2} = 0.6$, $s_{F1} = 30$, $s_{F2} = 10$). **(F)** A sketch showing how the switching probability, $p_{switch}(D_i)$, changes with the level of Delta in voxel $v_i$, $D_i$ (Eq (10)) with $p_{max} = 0.26$, $s_p = 0.0015$, $D_p = 1500$).

Its shape is characterised by two parameters: the mean value, $\mu$, and the distribution spread, $\kappa$ (see Fig 5C for a sketch of this pdf for different values of $\kappa$). $I_0$ is the modified Bessel function of the first kind of order 0.

In the context of EC migration, we view $\mu$ as the *mean polarisation angle*, and $\kappa$ as *cell exploratoriness*. We integrate ECM structure and composition and cell phenotype into our cell migration model by assuming that $\mu$ and $\kappa$ depend on these quantities. In particular, we assume that the mean polarisation angle, $\mu$, depends on the ECM fibril alignment which is represented in our model by the orientation landscape variable, $l_i$ (see Table 1). From a biological point of view, this is substantiated by experimental observations of cells forming focal adhesions with ECM fibrils and, consequently, aligning along them [43]. We introduce the *mean polarisation direction* vector, $\vec{p} \in \mathbb{R}^2$, and compute $\mu$ as its principle argument (see Fig 4)

$$\vec{p} = \left( \sum_{dir \in \mathcal{S}} H_{a,n}(l_i^{dir}) dir_x, \sum_{dir \in \mathcal{S}} H_{a,n}(l_i^{dir}) dir_y \right)^T,$$

$$\mu = Arg(\vec{p}).$$

(13)

In Eq (13) the summation is taken over all possible directions for movement, $dir = (dir_x, dir_y)^T \in \mathcal{S}$ (in a hexagonal lattice there are at most 6 possible directions, see Fig 5A). The Hill function, $H_{a,n}(\cdot)$, is used to reflect the natural saturation limit to alignment and deformation of the ECM fibrils, with $a$ and $n$ being fixed positive parameters. Details about how fibril orientation (orientation landscape, $l_i$) is calculated are given below.

Cell exploratoriness, $\kappa \geq 0$, is directly related to the EC phenotype (see Fig 1A I.). If $\kappa \approx 0$, then the effect of polarity on migration is weak, and cells can *explore* many directions (this behaviour is typical of exploratory tip cells, see Eq (3)). By contrast, when $\kappa \gg 1$, the von Mises pdf is concentrated around $\mu$ (this behaviour is characteristic of stalk cells, see Eq (3)). To account for such phenotype-dependent behaviour, we assume $\kappa$ to be a monotonic decreasing function of $D_i$, the Delta level of the migrating cell. Similarly, increased concentration of the BM components deposited by ECs (see Fig 1A IV.) reduces the exploratory capability of both tip and stalk cells. We therefore propose $\kappa$ to be an increasing function of the local concentration of the BM components, $m_i$. Combining these effects, we arrive at the following functional form for $\kappa = \kappa(D_i, m_i)$:

$$\kappa = \kappa(D_i, m_i) = K \exp (k_m m_i - k_D D_i).$$

(14)

Here $K, k_m, k_D$ are positive parameters (see Fig 5D for a sketch of $\kappa(D_i, m_i)$).

Since our model of cell migration is formulated on a lattice, transition rates of type $\omega(i \to j)$ (jumps from $v_i$ into $v_j$, in the direction $s \in \mathcal{S}$) are associated with an angle interval $[\phi_{min}^s, \phi_{max}^s]$ (see Fig 2B). This corresponds to the angle between the vectors connecting the centre of the voxel $v_i$ with the endpoints of the voxel edge shared by $v_i$ and $v_j$. For example, in a hexagonal lattice, the right direction, $r \in \mathcal{S}$, is associated with an interval $[\phi_{min}^r, \phi_{max}^r] = [-\pi/6, \pi/6]$. Therefore, we restrict the von Mises pdf, $f_{vM}(\cdot)$, in Eq (7) to this interval by multiplying it by a corresponding indicator function, $W^s(\phi)$,

$$W^s(\phi) = \begin{cases} 1, & \text{if } \phi + 2\pi k \in [\phi_{min}^s, \phi_{max}^s], \quad k \in \mathbb{Z}, \\ 0, & \text{otherwise.} \end{cases}$$

(15)

This function is $2\pi$-periodic due to the fact that its argument is an angle. We refer to $W^s(\phi)$ as the *window function* (see Fig 5B).

**Tissue scale. Modelling cell environment: ECM structure and composition.** We account for the alignment of ECM fibrils, $l$, density of the ECM, $c$, and concentration of the basal lamina components, $m$ (see Table 1) in the following way.

Local alignment of the ECM fibrils, $l$, serves as a scaffold for the orientated migration of ECs [43]. Thus, we refer to $l$ as the *orientation landscape*. Traction forces exerted by migrating ECs realign the ECM fibrils, so that they move closer to the growing sprouts. Furthermore, since tip cells have more filopodia than stalk cells, they exert a greater influence on the orientation landscape [39] (see Fig 1A V.). We account for phenotype-dependent ECM realignment by assuming active stretching and accumulation of the fibrils upon cell movement between the voxels $i \rightarrow j$ (in the direction $s \in \mathcal{S}$), i.e. when a transition of type $\omega(i \rightarrow j)$ occurs,

$$l_i^s = l_i^s + \Delta_l D_i,$$
$$l_j^s = l_j^s + \Delta_l D_i. \tag{16}$$

Here, the parameter $\Delta_l > 0$, which quantifies the linear response of ECM fibrils to cell migration, depends on the substrate stiffness.

Besides active stretching induced by cell locomotion, we also consider passive relaxation of the orientation landscape. We assume that relaxation follows a simple elastic model so that the orientation landscape decays exponentially at a constant rate, $\eta_l$,

$$l_i^s(t + \tau) = l_i^s(t) \exp\left(-\eta_l \tau\right), \tag{17}$$

where $\tau$ is the waiting time of the occurred migration transition and the update is done for all voxels $i \in \mathcal{I}$ and all directions $s \in \mathcal{S}$.

The time evolution of the ECM density, $c$, and BM components, $m$, is modelled via local ODEs. We assume phenotype-dependent ECM proteolysis induced by ECs (see Fig 1A III.). Since tip cells exhibit higher proteolytic activity than stalk cells [83, 84], we assume that the ECM at voxel $v_i$ is degraded at rate $\eta_c(D_i)$, which is an increasing function of its argument

$$\frac{dc_i}{dt} = \begin{cases} -\eta_c(D_i), & \text{if } c_i > 0, \\ 0, & \text{otherwise;} \end{cases} \tag{18}$$

$$\eta_c(D_i) = \frac{\eta_{max}}{1 + \exp\left(-s_c(D_i - D_c)\right)}. \tag{19}$$

Here, in order to account for the natural saturation in EC proteolytic ability, we assume a sigmoidal functional form for $\eta_c(D_i)$ with positive parameters $D_c$ and $s_c$ (which correspond to the threshold level of Delta for initiation of ECM proteolysis and sharpness of EC response, respectively) and maximum value $\eta_{max}$.

Similarly, BM assembly (i.e. deposition of basal lamina components), is assumed to be phenotype-dependent (see Fig 1A IV.). Tip cells are known to secrete BM components and to recruit and activate pericytes which secrete basal lamina components around the sprout [83, 84]. Thus we assume that the rate of secretion of BM components, $\gamma_m(D_i)$, is an increasing function of $D_i$

$$\frac{dm_i}{dt} = \begin{cases} \gamma_m(D_i), & \text{if } m_i < 1.0, \\ 0, & \text{otherwise.} \end{cases} \tag{20}$$

$$\gamma_m(D_i) \quad = \frac{\gamma_{max}}{1 + \exp\left(-s_m(D_i - D_m)\right)}, \tag{21}$$

where, again, a sigmoidal functional form is assumed for $\gamma_m(D_i)$ with positive parameters $D_m$ and $s_m$ (which correspond to the threshold level of Delta for initiation of BM assembly and sharpness of EC response, respectively) and maximum value $\gamma_{max}$. Here we assume the decay of BM components is negligible on the timescale of our simulations.

## Description of quantitative metrics

In order to calibrate and compare our simulation results to experimental data (see [2, 8]), we computed a number of metrics defined below.

**Displacement.** The displacement statistic is associated with the average distance travelled by a cell in 15 minutes [8].

**Orientation.** The orientation statistic measures the average persistence of cells as they move. It is computed as the average (over ECs in all performed realisations) quantity of ratios of the length of a smoothed trajectory to the actual trajectory travelled by a single cell during simulation [2].

**Directionality.** This metric measures the average proportions of cells moving in the direction of sprout elongation (anterograde), cells moving in the direction opposite to sprout elongation (retrograde), and cells that do not move during 20 minutes (still) [2].

**Tip cell proportion.** This metric is computed as the ratio of cells characterised by tip phenotype (see Eq (3)) to the total number of cells in the system at a given time point.

**Mixing measure.** This metric is motivated by the experimental observation that the trajectories of individual ECs, which initially form clusters with their immediate neighbours (see Fig 6 for an illustration), at later times diverge so that cells can find themselves at distant regions of the angiogenic vascular network [2, 8]. This metric is introduced to quantify the cell rearrangements. Cell rearrangements are a key driver of sprout elongation during the early stages of vasculature formation and, as such, are directly related to network growth patterns. Later in this work, we will show how the mixing measure varies for different patterns of vascular network formation.

Briefly, during simulations, we assign a label to each EC, $\iota$, and record its position in the system at time $t$, $p(\iota, t) \in \mathcal{I}$. We specify a set of voxels that form a cluster of nearest neighbours in the lattice, $\mathcal{I}_{cluster}$. At the end of the simulation, using recorded cell trajectories, we compute how far away (in a pair-wise fashion) cells that were situated at the voxels in $\mathcal{I}_{cluster}$ at time $t$ have moved away from each other during time interval of duration $t_m$. Normalising this quantity by the cardinal of $\mathcal{I}_{cluster}$, $|\mathcal{I}_{cluster}|$, and the maximum possible travel distance in the system, $d_{max}$, we obtain the *mixing measure* at time $t$, $\mathcal{M}(t)$, defined as

$$\mathcal{M}(t) = \frac{1}{|\mathcal{I}_{cluster}| \, d_{max}} \sum_{\substack{\iota_1, \ \iota_2 \text{ such that} \\ p(\iota_1, t) = i, \ p(\iota_2, t) = j \\ i, j \in \mathcal{I}_{cluster}, \ i \neq j}} \left( d(\iota_1, \iota_2, t + t_m) - d(\iota_1, \iota_2, t) \right). \tag{22}$$

Here $d(\iota_1, \iota_2, t)$ is the distance between cells with labels, $\iota_1$ and $\iota_2$, at time $t$. It is computed as a distance in a manifold of the simulated vascular network (see Fig 6). This is due to the fact that ECs do not migrate randomly but rather within ECM-free vascular guidance tunnels of the generated vascular network.

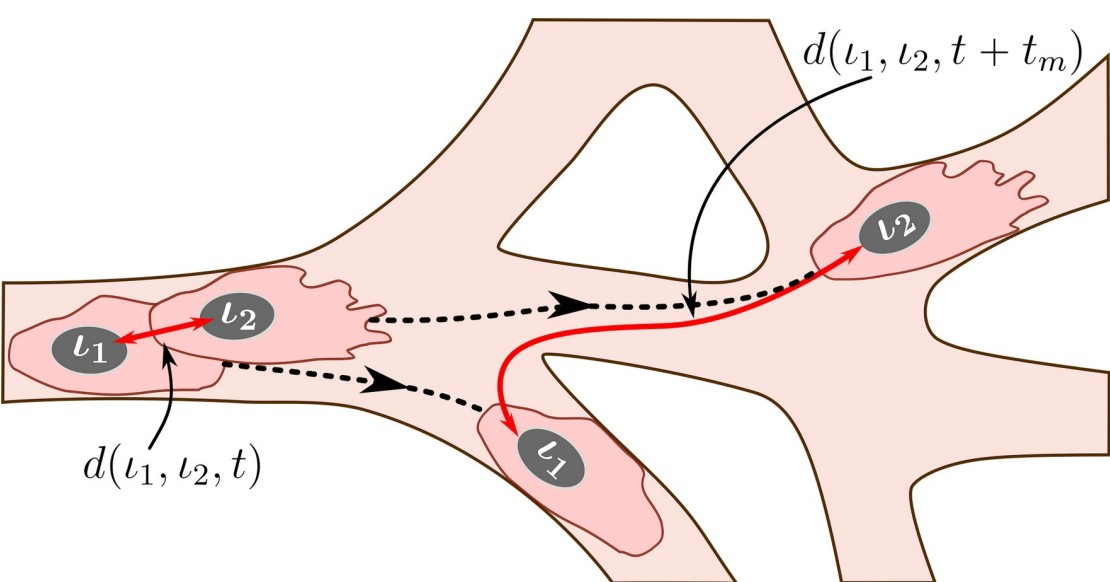

**Fig 6. An illustration of the mixing measure.** Two cells, labelled by $\iota_1$ and $\iota_2$, are located at the positions corresponding to the set $\mathcal{I}_{cluster}$ at time $t$. We track their trajectories in the simulated vascular network (dashed black lines) during time $t_m$. The mixing measure is defined as the difference between the distances between these cells at times $t$ and $t + t_m$, $d(\iota_1, \iota_2, t)$ and $d(\iota_1, \iota_2, t + t_m)$, respectively, normalised by the number of cells considered and the maximum distance it is possible to travel in the simulated network. The distance function is defined as a distance within the simulated network (see S1 Text for details).

The distance, $d_{max}$, in Eq (22), is the maximum distance in the simulated vascular network, defined as follows

$$d_{max} = \max_{\iota_1,\ \iota_2} d(\iota_1, \iota_2, T_{max}), \tag{23}$$

where $T_{max}$ is the final simulation time.

Detailed descriptions of all the metrics and computational algorithms that we used are given in S1 Text.

## Results

### Emergent qualitative features: Branching and VEGF sensitivity

Our model exhibits two characteristic features of functional angiogenic structures, namely, branching and chemotactic behaviour. A novel aspect of our multiscale model is that these features are emergent properties of its dynamics rather than being hardwired into the model.

Specifically, branching is a direct consequence of the phenotype-dependent polarity of individual cells. When a stalk cell within a sprout undergoes a phenotype switch and assumes a tip identity, its exploratoriness, increases (i.e. $\kappa$, decreases). This enables the cell to develop a polarity angle that departs from the mean elongation direction of the sprout, $\mu$. As a consequence, a new branch forms. In our model, new branches are typically initiated by cells exhibiting the tip cell phenotype (see S1 Movie). This behaviour is characteristic of ECs observed in biological experiments [7, 9]. Figs 7 and 8 illustrate the branching phenomenon and stabilisation of the network structure (due to accumulation of BM components deposited by ECs) in single realisations of numerical simulations of the model for uniform VEGF distribution at concentrations of 5 and 50 ng/ml (see also S1 and S4 Movies). Note that in all our simulations

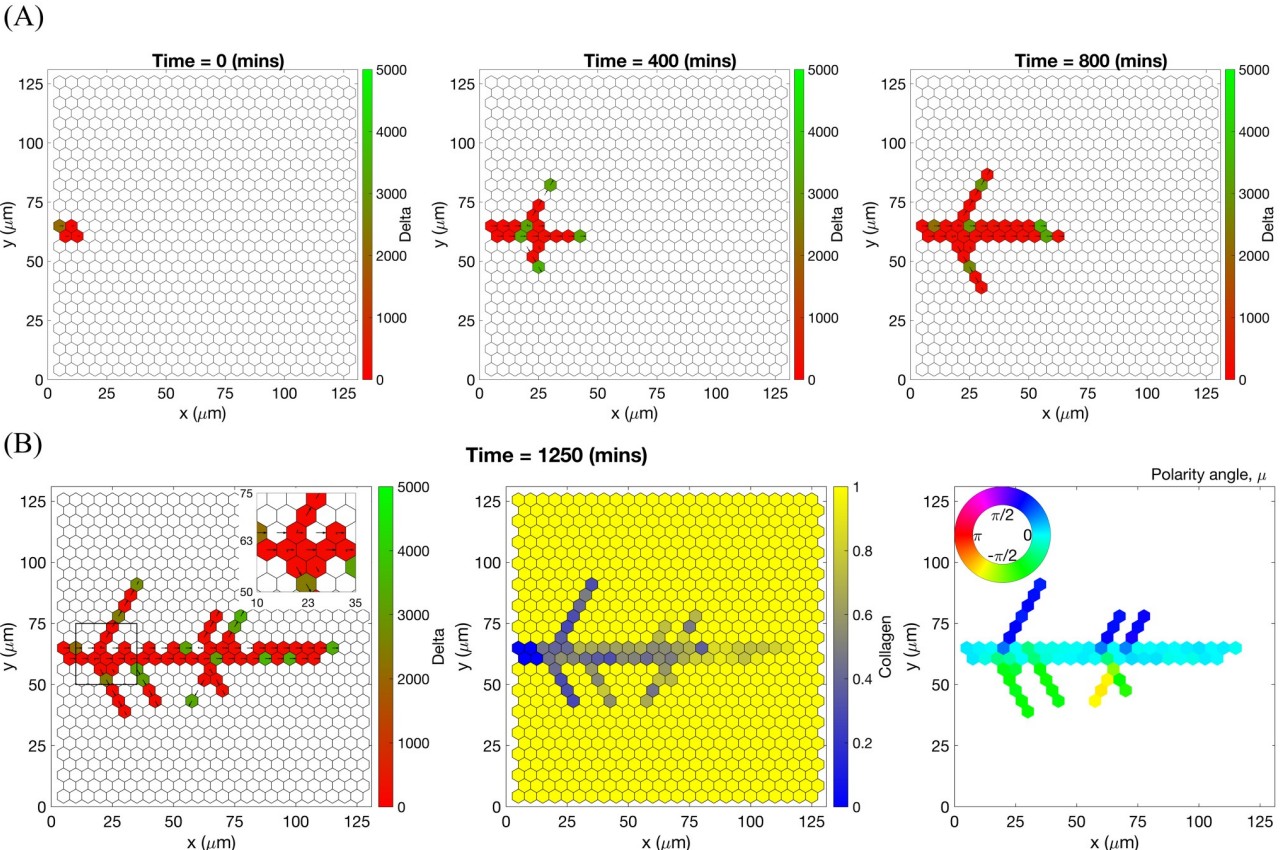

**Fig 7. An example of an individual vascular network generated during simulation of our model with uniform VEGF = 5 ng/ml. (A)** Temporal evolution of a simulated vascular network captured at 0 mins, 400 mins and 800 mins in an individual realization of our model for uniform VEGF = 5 ng/ml. The colour bar indicates level of Delta, **D**, (green colour corresponds to tip cells, red—to stalk cells). Arrows indicate the configuration of the orientation landscape, **l**. **(B)** The final configuration of the simulated vascular network at 1250 mins (corresponds to the final simulation time, $t = T_{max}$ = 2.5). The leftmost panel shows the concentration of Delta, **D**. Higher concentration (green colour) corresponds to tip cell phenotype, low concentration (red colour)—to stalk. On this plot, arrows correspond to the orientation landscape configuration, **l**. The central panel shows the final concentration of the ECM, **c**. The rightmost panel shows the final distribution of the polarity angle, **μ**, variable. Numerical simulation was performed using **Setup 1** from S4 Table. Parameter values are listed S1 and S2 Tables for subcellular and cellular/tissue scales, respectively. For a movie of the numerical simulation, see S4 Movie, left panel.

"gaps" within sprouts can arise since we track the positions of cell nuclei and do not account for their true spatial extent.

Chemotactic sensitivity in our model is a direct consequence of cell interactions with the ECM. In biological experiments, proteolytic activity of cells was observed to increase as expression levels of Delta rise [83, 84]. In our model, increased levels of extracellular VEGF, *V*, up-regulate subcellular levels of Delta. As a result, an EC's ability to degrade the ECM and invade it at a faster rate is enhanced where VEGF levels are high. This can be seen by comparing networks generated at different uniform VEGF concentrations (see Figs 7 and 8). The network generated at VEGF = 5 ng/ml is small (Fig 7B, leftmost plot), and the vascular guidance tunnels created via proteolysis (middle plot) are not fully formed. By contrast, the simulated network for VEGF = 50 ng/ml (Fig 8B, leftmost plot) has a greater spatial extent, since collagen-free vascular tunnels (Fig 8B, middle plot) facilitate cell migration within them, increasing sprouting and cell persistence (see *ECM density* term in Eq (7)).

Our model also exhibits the *brush-border* phenomenon [11, 85]. We performed a numerical simulation experiment of sprouting initialised from an initial vessel placed in a matrix with

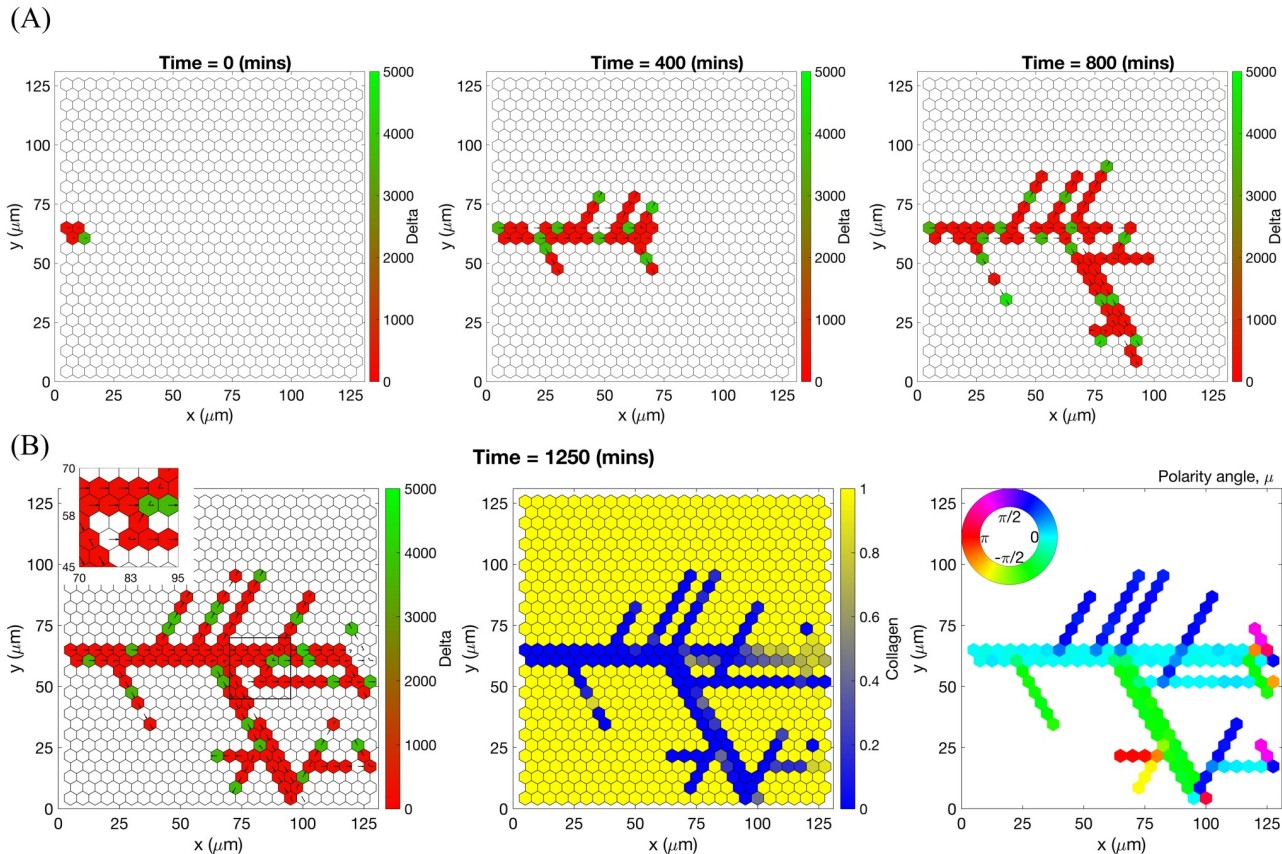

**Fig 8. An example of an individual vascular network generated during simulation of our model with uniform VEGF = 50 ng/ml. (A)** Temporal evolution of a simulated vascular network captured at 0 mins, 400 mins and 800 mins in an individual realization of our model for uniform VEGF = 50 ng/ml. The colour bar indicates level of Delta, $D$, (green colour corresponds to tip cells, red—to stalk cells). Arrows indicate the configuration of the orientation landscape, $l$. **(B)** The final configuration of the simulated vascular network at 1250 mins (corresponds to the final simulation time, $t = T_{max}$ = 2.5). The leftmost panel shows the concentration of Delta, $D$. Higher concentration (green colour) corresponds to tip cell phenotype, low concentration (red colour)—to stalk. On this plot, arrows correspond to the orientation landscape configuration, $l$. The central panel shows the final concentration of the ECM, $c$. The rightmost panel shows the final distribution of the polarity angle, $\mu$, variable. Numerical simulation was performed using **Setup 1** from S4 Table. Parameter values are listed in S1 and S2 Tables for subcellular and cellular/tissue scales, respectively. For a movie of the numerical simulation, see S1 Movie.

linearly increasing VEGF gradient. Fig 9 shows the evolution of the network at different times. The brush-border effect is evident at later times and characterised by increased cell numbers and branches in the top regions of the domain where VEGF levels are high.

## Model calibration

Having established that our model exhibits the essential features of branching and chemotactic behaviour observed in experiments, we next compared our simulations with experimental results from [2, 8, 38]. This enabled us to estimate baseline parameter values for processes at the cellular and tissue scales (estimated values of parameters associated with processes acting at the subcellular scale are taken from previous works, see S1 Table).

We ran 100 model simulations for uniform VEGF concentrations of 0, 5 and 50 ng/ml (**Setup 1**, see S4 Table, with final simulation time, $T_{max}$ = 2.5, equivalent to 1250 mins, i.e. $\approx$ 20.8 h) and computed three metrics, namely, *displacement*, *orientation* and *directionality* (see **Materials and methods** for definitions) as was done in [2, 8]. The results presented in

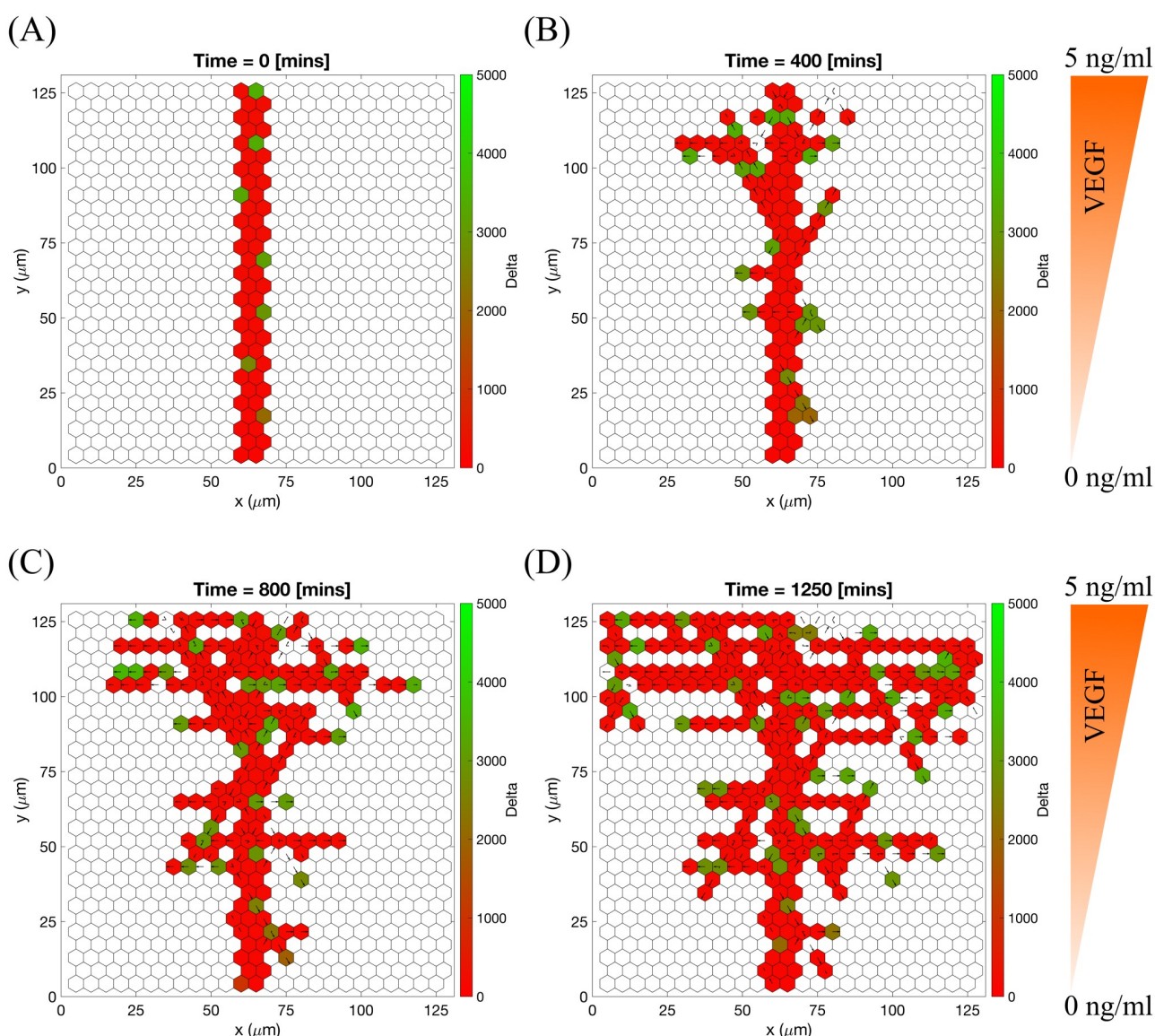

**Fig 9. Sprouting in static VEGF gradient.** Snapshots at different times of a vascular network growing in linear VEGF gradient increasing from VEGF = 0 ng/ml to VEGF = 5 ng/ml. **(A)** Time = 0 mins, initial setup. **(B)** Time = 400 mins, new branches appear from the initial sprout mostly in the upper half of the domain (higher VEGF concentration). ECs with lower positions have lower Delta level. **(C)** Time = 800 mins, the effect of the VEGF gradient can be seen clearly. **(D)** Time = 1250 mins, the final configuration of the simulated V-shaped (opening towards higher VEGF concentrations) network. Colour bar shows the level of Delta, **D**. Numerical simulation was performed using **Setup 2** from S4 Table with final simulation time, $T_{max}$ = 2.5. Parameter values are listed in S1 and S2 Tables for subcellular and cellular/tissue scales, respectively.

Fig 10 show that our simulations reproduce general trends of angiogenic sprouting reported in [2, 8]. Specifically, regarding the displacement statistic (Fig 10A), agreement is very good, except for the inconsistency at displacement = 0 $\mu m$, i.e. cells that did not move during the considered time interval (15 minutes). This inconsistency arises because we do not include the vascular bed from which cells migrate (we account for it via a boundary condition; see Fig 2A and S1 Appendix). By contrast, in [8], cell displacements from the embedded aortic ring assay were included in the sample (these ECs are mostly quiescent). Similarly, results regarding the orientation statistic (Fig 10B and 10C) are in good agreement with the experimental results. In

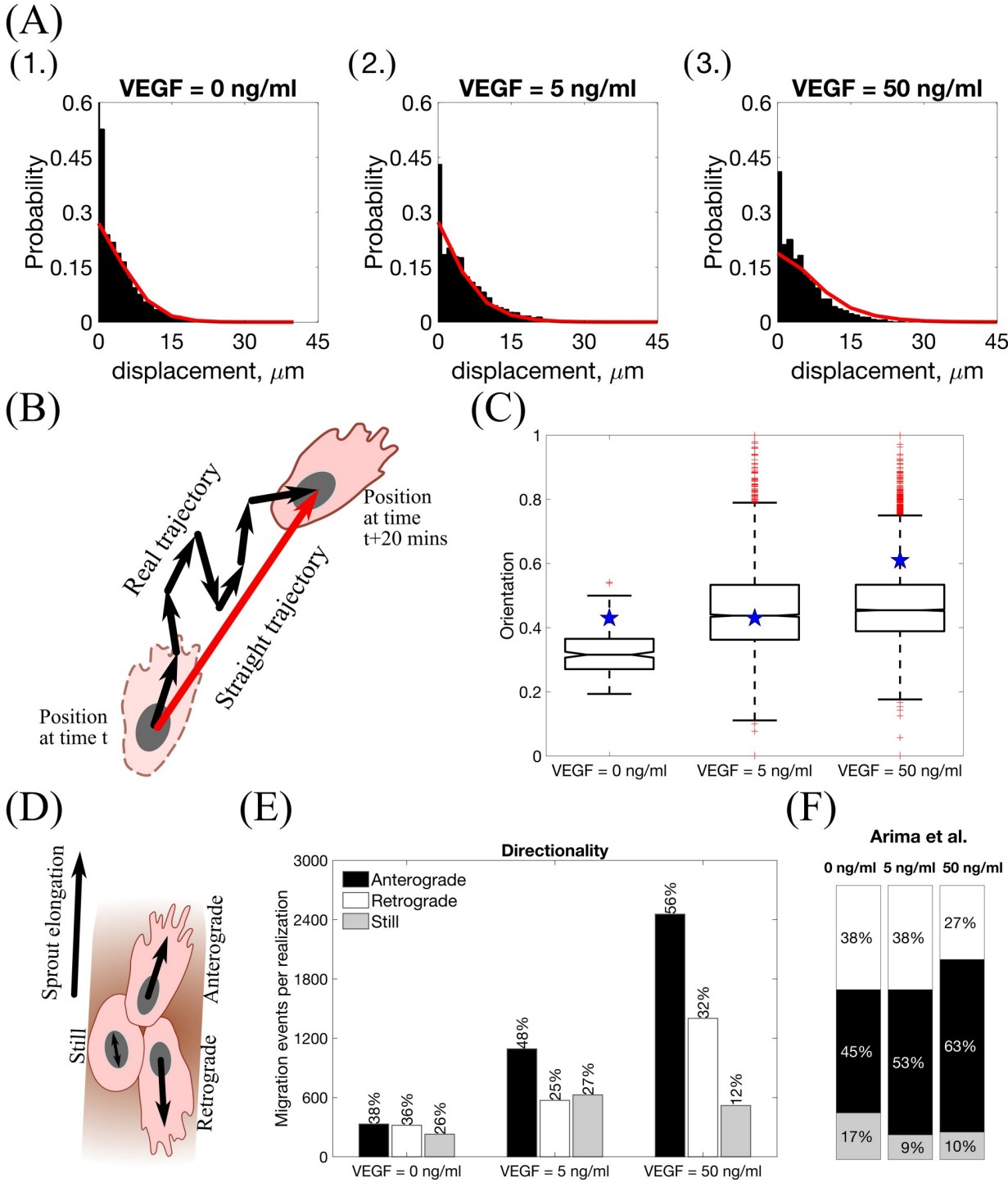

**Fig 10. Statistics extracted from simulations of our model.** **(A)** Histograms of cell displacements during a 15 minute time period for **(1.)** VEGF = 0 ng/ml, **(2.)** VEGF = 5 ng/ml and **(3.)** VEGF = 50 ng/ml. Black histograms correspond to the experimental data taken from the Supplementary Material of [8], red lines correspond to displacement curves for each VEGF concentration extracted from our model simulations. **(B)** A cartoon illustrating the orientation statistic, which is defined as a ratio between the net trajectory and the actual trajectory of a cell during simulation. **(C)** Box plots of the orientation statistic extracted from model simulations with VEGF = 0, 5, 50 ng/ml. Red crosses indicate box plot outliers. Orientation statistics obtained from experimental data from [2] are shown by blue stars on each box plot. **(D)** A cartoon illustrating the directionality statistic. **(E)** The directionality statistics for model simulations with VEGF = 0, 5, 50 ng/ml. **(F)** The directionality statistics extracted from experimental data in [2]. Numerical simulations were performed using **Setup 1** from S4 Table and $T_{max}$ = 2.5. Parameter values are listed in S1 and S2 Tables for subcellular and cellular/tissue scales, respectively. All statistics were computed for 100 realisations.

particular, in our model ECs are more oriented, i.e. more persistent, in higher VEGF concentrations, which is a feature also observed in [2]. Concerning the directionality statistic (Fig 10D–10F), we note that when VEGF = 0 ng/ml the numbers of anterograde and retrograde cells in the experiments [2] (Fig 10F) and numerical simulations (Fig 10E) are approximately equal. In this scenario, ECM proteolysis is slow and cell migration is mostly constrained to existing sprouts. Thus, any anterograde movement is an overtaking event in which the overtaken cell has to perform a retrograde displacement. As the VEGF concentration increases, ECM proteolysis (see Eqs (18) and (19)) increases and more cells at the leading edge of sprouts can invade the surrounding ECM, elongating the sprouts. This leads to an increase in the ratio of anterograde to retrograde moving cells with the VEGF concentration.

Further evidence of agreement with experimental data is found by performing numerical simulations imitating the experimental setup of [38]. We performed simulations of sprouting from a cell bead embedded into the ECM (see **Setup 3** from S4 Table) with varying collagen density (which corresponds to the initial ECM concentration, $c_{max}$, in our model) and a static linear VEGF gradient. Results from single realisations of different values of $c_{max}$ are presented in Fig 11 (see also S2 Movie). These results show free cell migration with no preferred direction for low $c_{max}$ values, typical angiogenic morphology for intermediate values of $c_{max}$, and poorly elongating sprouts for higher values of $c_{max}$. These findings are consistent with the experimental observations reported in [38]. Furthermore, we note that the "sweet spot" of ECM concentration is related to EC ability to form typical angiogenic sprouting structures rather than to their ability to invade the ECM which decreases as the ECM concentration, $c_{max}$, increases (see Fig 11).

We note that the results presented in this section were generated using a fixed set of parameter values, except for the concentration of VEGF, $V$, and the concentration of collagen, $c_{max}$. Henceforth, we use these values as baseline parameter values (see S2 Table).

## Further model validation

In this section we validate our model by comparing its predictions with experimental results detailing the behaviour of certain VEGF receptor mutant cells (VEGFR2$^{+/-}$ and VEGFR1$^{+/-}$ mutants with halved gene expressions of VEGFR2 and VEGFR1, respectively), studied by Jakobsson et al. in [7] and described in S1 Appendix.

In order to ascertain whether our model can quantitatively reproduce competition between cells of different lineages (wild type (WT) and mutant cells) for the position of the leading cell in a sprout, we designed a series of numerical experiments which mimic the biological experiments reported in [7]. We start by simulating of EC competition within linear sprouts that are devoid of collagen matrix (**Setup 4** in S4 Table), to ensure proteolysis-free random shuffling of cells within the sprout. We randomly initialise the sprout with cells of two chosen types with probability 50% (50% of WT cells and 50% of a specific mutant cell type) (see Fig 12, left column). Cells are then allowed to shuffle within the sprout, overtaking each other. For each realisation we record the total amount of time for which WT and mutant cells occupy the position of the leading cell.

As a control, we ran simulations in which two identical cell lines with parameters corresponding to WT lineage were mixed in a 1:1 ratio. As expected, the contribution of each WT cell to the leading cell position was approximately 50% (see Table 3).

We then performed competition simulations in which WT cells were mixed with the different mutant cell lines in a 1:1 ratio (i.e. mixing 50% of WT cells with 50% of mutant cells). We repeated these numerical experiments in the presence of DAPT inhibitor which abolishes Notch signalling. The results of individual realisations presented in Fig 12 (see also S3 Movie),

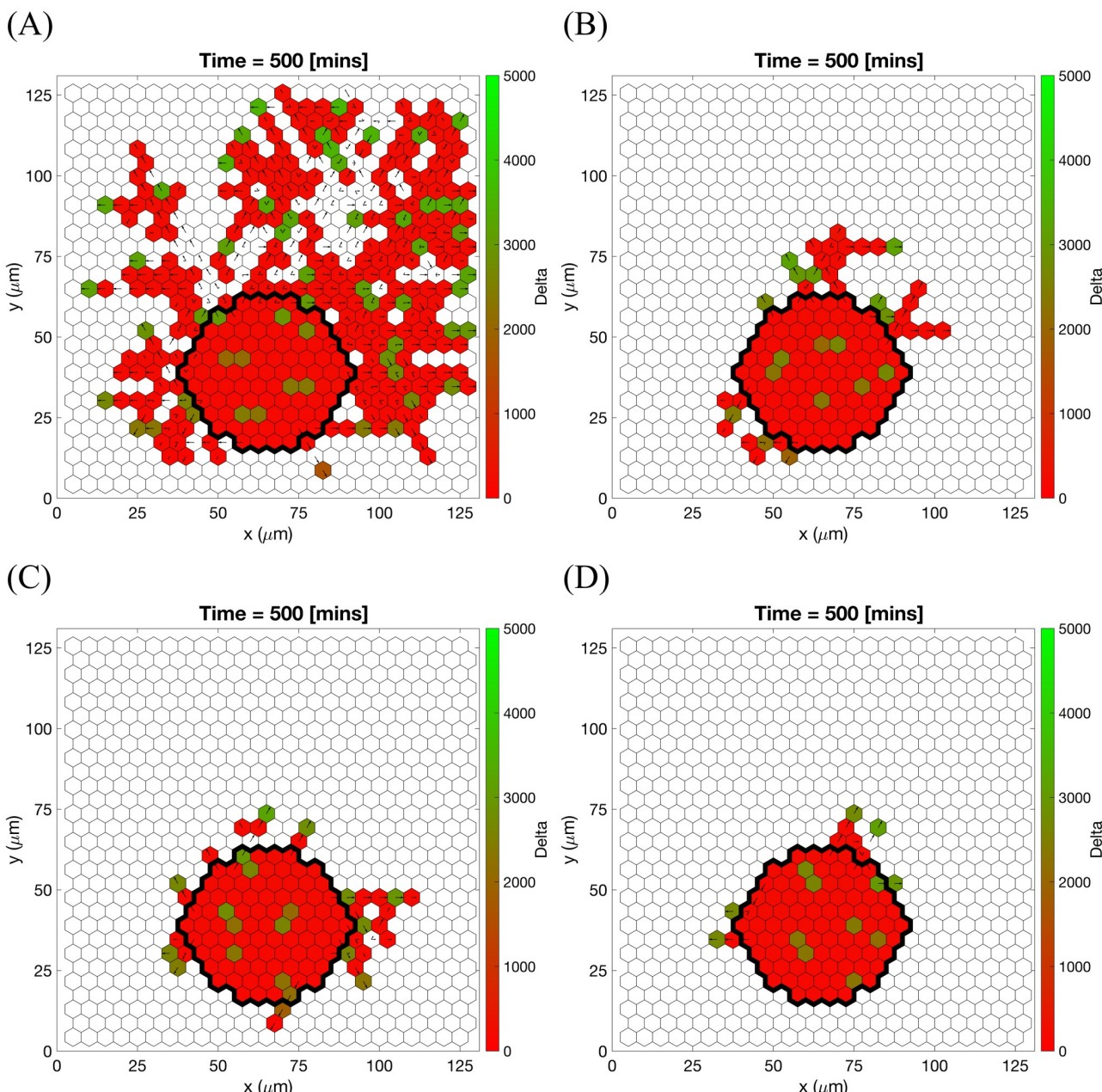

**Fig 11. Cell migration from a cell bead in substrates of different collagen density.** Final configurations of simulated vascular networks at time $T_{max} =$ 1.0 (corresponding to 500 minutes) of individual simulations used in reproducing the results of the polarisation experiment in [38]. Maximum collagen density **(A)** $c_{max} = 0.1$, **(B)** $c_{max} = 1.0$, **(C)** $c_{max} = 1.7$, **(D)** $c_{max} = 3.0$. The VEGF linear gradient starts with 0 ng/ml at $y = 0$ and increases up to 5 ng/ml at $y = 125$ $\mu m$. Central bead initial and basement membrane conditions, $\mathcal{I}_{BM} = \mathcal{I}_{init}$, are outlined by a black thick line on each plot. Colour bars indicate Delta ligand concentration. Numerical simulations were performed using **Setup 3** from S4 Table. Parameter values are listed in S1 and S2 Tables for subcellular and cellular/tissue scales, respectively. For a movie of the numerical simulation, see S2 Movie.

illustrate features of the different competition scenarios. For the WT:VEGFR2$^{+/-}$ scenario (see Fig 12A) when the mutant cells compete with WT cells, they almost never acquire the tip phenotype. Consequently, they are rapidly overtaken by WT cells and accumulate far from the leading edge of the sprout (outlined in cyan on each plot). The leading cell positions are thus occupied predominately by WT cells. By contrast, in the WT:VEGFR1$^{+/-}$ scenario (Fig 12B),

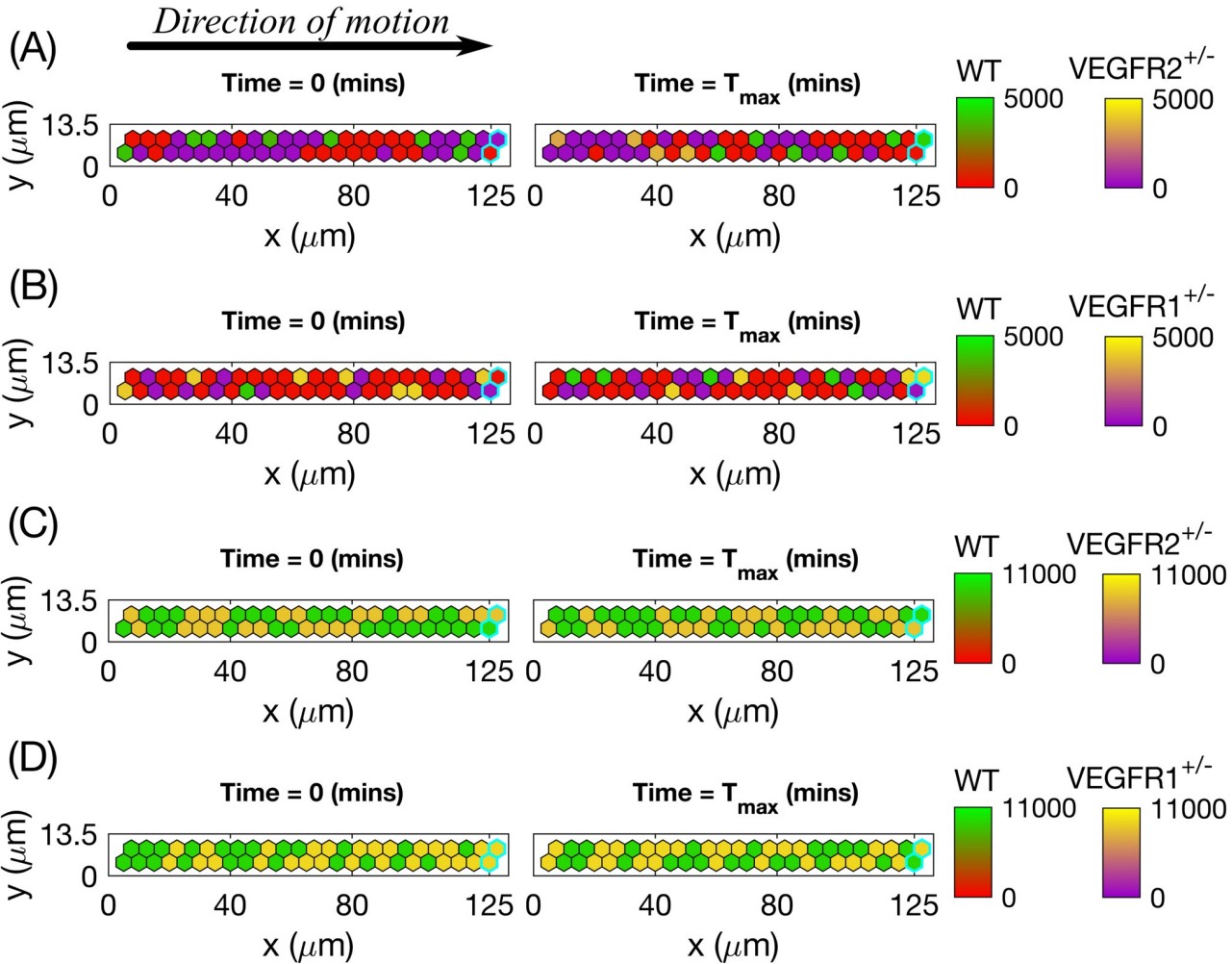

**Fig 12. Initial and final configurations of single realisations of cells shuffling within a linear sprout when two given cell lines are mixed 1:1 (50% to 50%).** Left column corresponds to the initial (random, 1:1) distribution of cells, right column—to the final one. The colour bar for Delta level of the WT goes from red colour (stalk cell) to green (tip cell), whereas for the mutant cells the bar goes from purple colour (stalk cell) to yellow (tip cell). **(A)** 50% of WT cells mixed with 50% of VEGFR2$^{+/-}$ mutant cells, no DAPT treatment. **(B)** 50% of WT cells mixed with 50% of VEGFR1$^{+/-}$ mutant cells, no DAPT treatment. **(C)** 50% of WT cells mixed with 50% of VEGFR2$^{+/-}$ mutant cells, both DAPT-treated. **(D)** 50% of WT cells mixed with 50% of VEGFR1$^{+/-}$ mutant cells, both DAPT-treated. Voxels corresponding to the leading edge of a sprout are outlined by thick cyan lines on each plot. Numerical simulations were performed using **Setup 4** from S4 Table. Parameter values are listed in S1 and S2 Tables for subcellular and cellular/tissue scales, respectively, except for the changed parameters for the mutant cells listed in S1 Appendix. Final simulation time, $T_{max}$ = 50.0. For a movie of the numerical simulations, see S3 Movie.

**Table 3. Contribution of WT cells to the leading cell position when mixed 1:1 (50%: 50%) with another type of cell (equivalent WT or specified mutant).** Since as an initial setup of simulations we considered a sprout of width 2 (two voxels), there are two equivalent leading positions (Position 1 and Position 2) (outlined on each plot by cyan lines in Fig 12). The results are reported as mean value ± standard deviation for samples obtained from 100 realizations for each experimental scenario. Numerical setup is as specified in Fig 12.

| Experiment | Position 1 | Position 2 | Ref. value [7] |
|---|---|---|---|
| WT:WT | 51.1±16.4% | 53.3±20.6% | 45.8% |
| WT:VEGFR2$^{+/-}$ | 93.6±7.1% | 90.4±14.8% | 87.0% |
| WT:VEGFR1$^{+/-}$ | 19.5±9.8% | 20.5±17.8% | 30.0% |
| WT:VEGFR2$^{+/-}$ +DAPT | 51.5±13.7% | 52.9±17.8% | 47.0% |
| WT:VEGFR1$^{+/-}$ +DAPT | 50.3±14.3% | 47.6±16.7% | 40.6% |

mutant cells acquire the tip phenotype more often than WT cells, and thus contribute more significantly to the leading cell position. Treatment with DAPT forces all ECs (WT, VEGFR2$^{+/-}$ and VEGFR1$^{+/-}$) to acquire the tip cell phenotype (which is the default when Notch signalling is abolished [12, 13]). Consequently, in these cases both treated mutants have a 50% likelihood of occupancy of the leading position (Fig 12C and 12D).

We have also collected statistics from 100 realisations for each scenario and compared the results to the quantitative estimates provided in [7]. The results reported in Table 3 show that the contribution of WT cells to the leading cell position in each scenario is in good agreement with experimental values from [7]. In our simulations, VEGFR2$^{+/-}$ cells are less likely to stay at the leading position than WT cells: they occupy the leading position approximately 7% of the time. By contrast, VEGFR1$^{+/-}$ cells occupy the leading cell position approximately 78% of the time. DAPT restored the balance between the cells of different lineages so that they were on average equally mixed. Since the only parameters that we have modified are those used to mimic mutant cell gene expression and DAPT inhibition of Notch signalling (see S1 Appendix) our model provides possible explanation for overtaking dynamics of ECs in angiogenesis.

## Sensitivity analysis

To ascertain how variation in the baseline parameter values of our model affects the behaviour of the system, we have performed an extensive sensitivity analysis. Since the subcellular VEGF-Delta-Notch model has already been calibrated and validated independently [76], we have focused our analysis on the cellular and tissue scale parameters (see S2 Table). Briefly, we have performed our analysis by fixing all the parameters except one at their baseline values, and then vary the focal parameter by ±0.1%, ±5%, ±10%, ±15%, and ±20%. This procedure is repeated for each of the tissue and cellular scale parameters. In order to quantify the impact of the variation of each parameter on both EC behaviour and network structure we have measured the following quantities:

- anterograde cell proportion (directionality metric);

- orientation;

- displacements;

- number of branching points per 100 $\mu m^2$ of vascular network area;

- number of vessel segments;

- vessel segment lengths.

Each of these metrics has been measured over 100 realisations of the multiscale system. For a full account of the details, see S1 Text.

The results of our sensitivity analysis are summarised in Fig 13 and S2 Fig. Our analysis shows that system behaviour is robust to variations in most of the model parameters considered. This is indicated by the central cluster in Fig 13 and S2 Fig, highlighted in magenta, which represents those scenarios that exhibit very small deviations from the baseline behaviour. By contrast, variations of a small number of parameters produce significant deviation from the baseline behaviour (see Table 4 and Fig 13 and S2 Fig).

Specifically, we observe that an increase in $D_m$ and a decrease in both $D_c$ and $K$ induce excessive branching, with shorter average vessel length (see Fig 13, hyper-branching region highlighted in brown). By contrast, a decrease in $D_m$ and an increase in both $D_c$ and $K$ induce less branched networks, with longer average vessel length (see Fig 13, hypo-branching region highlighted in grey). These results are in agreement with well-known features of tumour

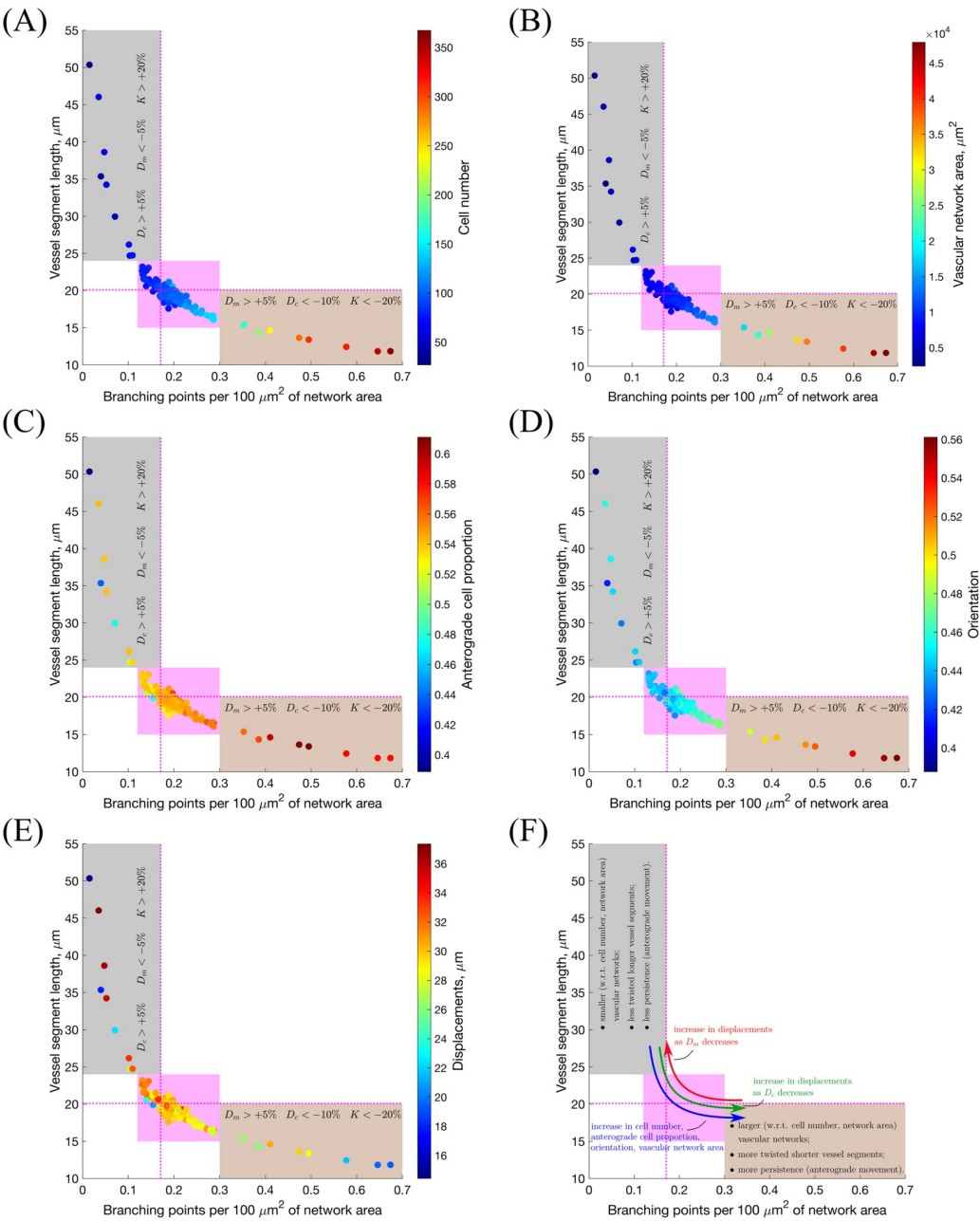

**Fig 13. Sensitivity analysis: Branching vs. vessel elongation.** The results of the sensitivity analysis are represented as scatter plots of the mean number of branching points per 100 $\mu m^2$ of vascular network area vs. mean vessel segment length, with colouring indicating mean **(A)** cell number; **(B)** vascular network area; **(C)** anterograde cell proportion; **(D)** orientation and **(E)** displacements. On these plots, dashed magenta lines indicate the point corresponding to the default parameter values (see S2 Table); magenta highlights the region of the main point clustering. The grey-coloured outlier region corresponds to vascular networks with few branching points and long vessel segments (hypo-branching), whereas the brown outlier region is characterised by short vessel segments and greater number of branching points (hyper-branching). Variations of the parameters that push the system towards one of the outlier regions are indicated on each plot. Panel **(F)** provides a general summary of these results. Simulation setup as in **Setup 1**, S3 Table, with $T_{max}$ = 2.5. The results are averaged over 100 realisations for each scenario. The subcellular parameters are fixed at their default values in all experiments (see S1 Table).

**Table 4. Sensitivity analysis: Parameters producing significant deviation in system behaviour from the baseline scenario.** ↓ stands for decrease of the focal parameter, ↑—increase of the focal parameter.

| Par. | Description | Ref. equation | Metrics affected | Effect |
|---|---|---|---|---|
| $D_c$ | Threshold of level of Delta for initiation of ECM proteolysis | Eq (19) | All | Hyper-branching: $D_c\downarrow$; hypo-branching: $D_c\uparrow$ |
| $D_m$ | Threshold of level of Delta for initiation of BM assembly | Eq (21) | All | Hypo-branching: $D_m\downarrow$; hyper-branching: $D_m\uparrow$ |
| $K$ | Cell exploratoriness, i.e. controls the variance of the von Mises distribution | Eq (14) | All | Hyper-branching: $K\downarrow$; hypo-branching: $K\uparrow$ |
| $E_{F1}$ | Threshold of the level of cell-cell contact necessary for cell movement initiation | Eq (9) | Orientation, directionality | Hyper-branching: $E_{F1}\downarrow$ |
| $E_{F2}$ | Threshold of the inhibitory effect of crowding on cell movement | Eq (9) | Orientation, directionality | Hypo-branching: $E_{F2}\downarrow$ |

vasculature, where the tumour microenvironment inhibits vessel stabilisation; specifically, it hinders the formation of the basal membrane in tumour vasculature, yielding aberrant, excessively branched, networks [86]. This phenomenon can be realised by increasing $D_m$. Furthermore, proteolysis is also up-regulated during tumour-induced angiogenesis due to secretion of MMPs by cancer cells. Proteolysis reduces the resistance experienced by the ECs as they migrate towards the tumour [86]. This effect can be accounted for phenomenologically by a reduction in $D_c$. Increased stimulation with growth factors that can also bind to VEGF receptors (as in pathological angiogenesis [86, 87]) can reduce the response of the ECs to chemotactic stimuli, due to high occupancy of receptors all over the cell membrane, thus shifting cell behaviour to chemokinesis (non-directional cell migration) [88, 89]. In our model, this transition is controlled by the cell exploratoriness, $\kappa$ (see Eq (14)): for high values of $\kappa$ (i.e. higher values of $K$) cell migration is directed along the sprout elongation vector, whereas for small values of $\kappa$ (i.e. smaller values of $K$) cells exhibit exploratory behaviour corresponding to chemokinesis.

Our model thus predicts that changes in $D_c$, $D_m$ and $K$ are likely to occur in tumour-induced angiogenesis. This prediction is supported by current knowledge regarding the effects of the presence of a tumour in the microenvironment [86–88].

## Model predictions: Network structure and cell mixing

We also simulated the growth of vascular networks formed by a single mutant cell line (VEGFR2$^{+/-}$ or VEGFR1$^{+/-}$) in the presence/absence of DAPT and compared our findings with the results from Jakobsson et al. who observed that in the absence of DAPT mutant cells mix with WT cells to form normal networks [7]. By contrast, the addition of DAPT leads to unstructured growth [7]. This is consistent with our simulation results (see S3–S6 Figs). Specifically, simulations with VEGFR2$^{+/-}$ mutant cells in the absence of DAPT (see S3 Fig and S4 Movie) suggest that the rate of network growth of VEGFR2$^{+/-}$ is slower than for their WT counterparts (see Figs 7 and 8 and S7 Fig). Since Delta levels in VEGFR2$^{+/-}$ cells are lower than in WT cells, they are less able to degrade and invade ECM, and deposit BM components than WT cells. This results in slower sprout elongation and increased branching (see S7 Fig). By contrast, VEGFR1$^{+/-}$ mutant cells possess higher levels of Delta and thus degrade the ECM more efficiently and invade the matrix more quickly than the WT cells (see S5 Fig and S4 Movie). Likewise, since the rate of segregation of BM components, $\gamma_m$, increases with Delta (Eq (21)), VEGFR1$^{+/-}$ cells are more persistent than WT cells. As a result, VEGFR1$^{+/-}$ ECs form less branched networks, with longer sprouts (see S7 Fig). Regarding networks grown with DAPT, since DAPT treatment abolishes all Notch signalling, all cells in the simulations

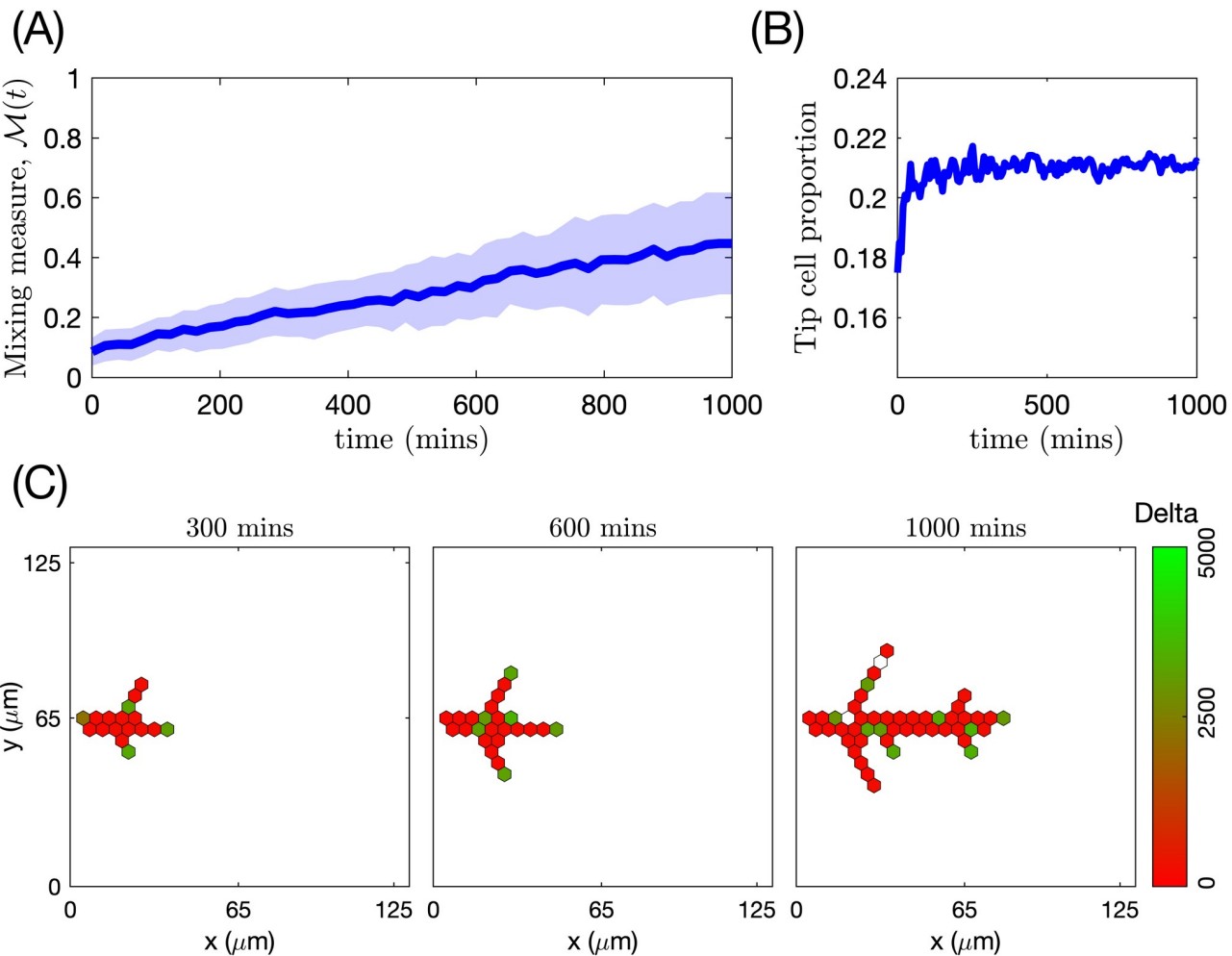

**Fig 14. Temporal evolution of mixing measure, tip cell proportion and branching structure in a simulated vascular network formed by WT cells.**
**(A)** The mixing measure, $M(t)$, as a function of time (the mean value is indicated by a thick line and standard deviation is shown by a colour band). The results are averaged over 100 realisations. **(B)** Evolution of tip cell proportion as a function of time. The results are averaged over 100 realisations. **(C)** Snapshots from a single realization of our model simulating a vascular network formed by WT cells at 300, 600 and 1000 minutes. Colour bar indicates the level of Delta. The numerical simulation setup used is **Setup 1** from S4 Table with final simulation time $T_{max} = 2.5$. VEGF distribution was fixed uniformly at 5 ng/ml. Parameter values are listed in S1 and S2 Tables for subcellular and cellular/tissue scales, respectively.

with DAPT acquire the tip cell phenotype (S5 and S6 Figs), which produces unstructured growth.

Since a cells' ability to compete for the leading position, or, equivalently, cell shuffling, is altered in mutant cells, we sought to understand how cell rearrangements influence the structure of a growing vascular network. To quantify cell rearrangements, we introduce a metric, which we refer to as *mixing measure*, $M(t)$ (see Eq (22)). In Fig 14A, we plot the dynamics of the mixing measure, $M(t)$, obtained by averaging over 100 WT simulations. As the vascular network grows and new sprouts form (see Fig 14C), $M(t)$ increases over time.

The time evolution of the mixing measure varies for different cell lines. For VEGFR2$^{+/-}$ mutant cells it increases more slowly (Fig 15A), than for the WT cells (Fig 14A). VEGFR2$^{+/-}$ mixing arises more from branching than sprout elongation (see Fig 15C). A similar trend is seen for the VEGFR1$^{+/-}$ cells (compared to WT cells) (Fig 15A). However, contrary to VEGFR2$^{+/-}$, this is due to high migration persistence of VEGFR1$^{+/-}$ ECs (see Fig 15D). A

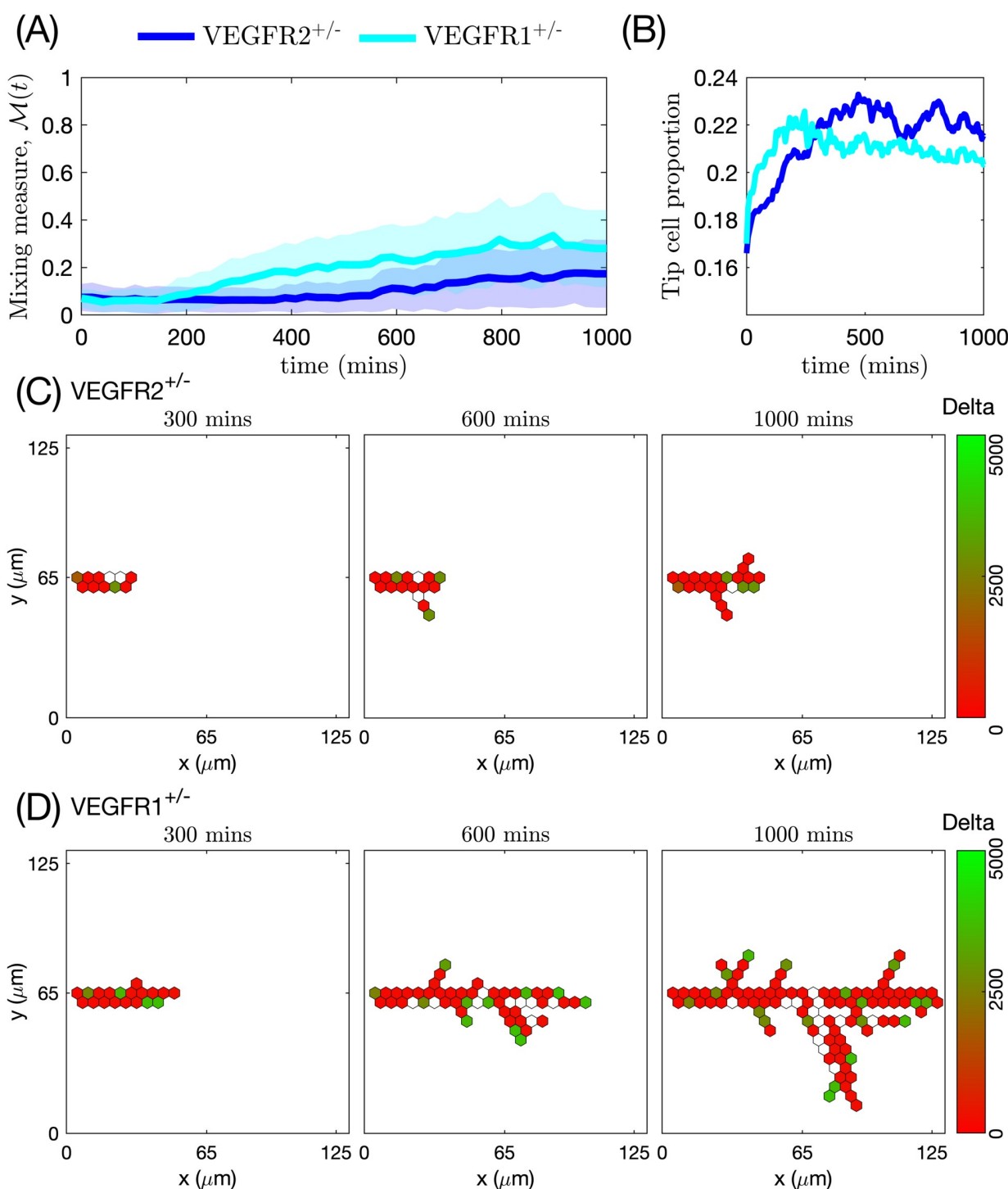

**Fig 15. Temporal evolution of mixing measure, tip cell proportion and branching structure in simulated vascular networks formed by VEGFR2$^{+/-}$and VEGFR1$^{+/-}$ mutant cells. (A)** The mixing measure, $M(t)$, as a function of time (the mean value is indicated by a thick line and standard deviation is shown by a band of the corresponding colour). The results are averaged over 100 realisations. **(B)** Evolution of tip cell proportion as a function of time. The results are averaged over 100 realisations. **(C)** Snapshots from a single realization of our model simulating a vascular network formed by VEGFR2$^{+/-}$ mutant cells at 300, 600 and 1000 minutes. Colour bar indicates the level of Delta. **(D)** Snapshots from a single realization of our model simulating a vascular network formed by VEGFR1$^{+/-}$ mutant cells at 300, 600 and 1000 minutes. Colour bar indicates the level of Delta. The numerical simulation setup used is **Setup 1** from S4 Table with final simulation time $T_{max}$ = 2.5. VEGF distribution was fixed uniformly at 5 ng/ml. Parameter values are listed in S1 and S2 Tables for subcellular and cellular/tissue scales, respectively, except of those changed for mutant cells (see S1 Appendix).

slower increase in the mixing measure for mutant cells correlates with slower stabilisation of tip cell proportion around its steady state in vascular networks formed by mutant cells (see Figs 14B and 15B for WT and mutant cells, respectively). This supports our hypothesis that cell shuffling is directly related to the phenotypic specifications of ECs. Thus, we note that the cell rearrangement phenomenon is not a cell-autonomous decision but rather a result of contact-dependent EC cross-talk, which leads to EC phenotype specification.

In all cases, regardless of the cell type and VEGF concentration, the mixing measure increases towards a steady state value (see S8 Fig, left panels). Furthermore, the steady state value of the mixing measure is independent of cell type ($\mathcal{M}(t) \approx 0.385$ as $t \to \infty$, see S9 Fig). This is a result of the fact that, as the vascular network reaches a sufficient size, cells perform proteolysis-free random shuffling within the manifold of the developed sprouts [52]. The rate of this proteolysis-free random shuffling does not depend on the cell line but rather on the tip-to-stalk ratio in the vasculature, which evolves to a steady state regardless of cell type and VEGF concentration (see S8 Fig, right panels). We conclude that the temporal evolution of the mixing measure characterises the resulting network structure to a larger degree than its steady state.

## Discussion and conclusions

Angiogenic sprouting has been extensively investigated from both experimental [1–9] and theoretical [16–33] perspectives. Integrating the variety of approaches and results from all these fields has allowed researchers to disentangle to a great extent mechanisms of complex EC behaviour such as coordinated EC migration [3, 7, 14] and EC-ECM interactions [33]. Nonetheless, as new experimental results emerge, biological assumptions of mathematical and computational models must be revisited, the models validated with new experimental data and used to test new hypotheses and generate predictions. As such, this multidisciplinary effort becomes a potential tool in designing biological experiments (e.g. for identifying new drug targets [90, 91], finding new mechanisms for abnormal cell behaviour [92], etc.).

Recently, it has become clear that cell rearrangements play a key role in driving vascular network growth in angiogenesis [1, 2, 7]. Defects in cell rearrangements lead to anastomosis failure and smaller vascular networks characterised by superimposed aberrant layers [1]. However, the functional role of this phenomenon remains to be explained [1, 2, 7]. Quantifying cell rearrangements and relating them to vascular patterning for ECs with varying gene expression patterns could be a starting point for understanding the functionality of cell mixing in angiogenesis.

In this work, we developed a multiscale model which integrates individual cell gene expression, EC migration and interaction with the local ECM environment. Our model exhibits characteristic EC behaviour, such as branching and chemotactic sensitivity, as emergent properties instead of being encoded via *ad hoc* rules (as has been traditionally done in the literature). The vascular networks generated by our model are capable of reproducing the general traits of sprouting angiogenesis: the networks exhibit branching patterns (see Figs 7 and 8), sprout elongation is enhanced in higher VEGF stimulation (see Fig 8) and the brush-border effect can be observed in networks grown in VEGF gradients (see Fig 9). Our simulation results are in good quantitative agreement with the characteristic trends of angiogenesis observed in experiments [2, 7, 8, 38] (see Figs 10 and 11 and Table 3).

We then used our model to quantify the phenomenon of cell rearrangement. We defined and introduced a mixing measure, $\mathcal{M}(t)$, (see Fig 6), for networks formed by WT cells and VEGFR2$^{+/-}$ and VEGFR1$^{+/-}$ mutant cells with impaired gene expression of VEGFR2 and VEGFR1, respectively, used in [7] (see Figs 14 and 15).

In all cases, in agreement with experimental observations, the mixing measure increases over time, although the specific details of its temporal evolution vary for different cell lines (see S8 Fig). In particular, for mutant cells, we find that mixing is lower due to either poor sprout elongation (VEGFR2$^{+/-}$ lineage, see Fig 15A and 15C and S7 Fig) or elevated cell persistency (VEGFR1$^{+/-}$ lineage, see Fig 15A and 15D and S7 Fig). WT cells form more functional networks, in the sense of more effective coverage of the domain (and thus future delivery of oxygen/nutrients). This is achieved by a balance between branching and sprout elongation which increases the mixing measure for WT cells. We thus showed that the time evolution of the mixing measure is directly correlated to the generic features of the vascular pattern. This result supports the claim that shuffling and cell mixing are essential for network formation and structure.

We also observe that the mixing measure reaches a steady state (see S9 Fig). We hypothesise that this is directly related to the proportion of tip cells in the network since they are the main driver of cell overtaking. This is substantiated by our results that, while the gene expression of VEGFR1$^{+/-}$ and VEGFR2$^{+/-}$ mutant cells exhibits variations, the steady state of the tip cell proportion is the same for all cell lineages (see S8 Fig, right column). Thus, although the branching pattern and effective sprout elongation vary for mutant cells, they generate adequate vascular networks in our simulations (see S3 and S4 Figs). We suggest that pathological network formation is directly related to the imbalance in the tip cell proportion (for example, treatment with DAPT, which abolishes Notch signalling and forces all ECs to acquire tip phenotype, leads to hyper-sprouting [7], see S5 and S6 Figs). Analysing this proportion and what triggers its change might help to understand better what leads to malformations in sprouting angiogenesis. Furthermore, the results of our sensitivity analysis suggest that variations in the parameters that control ECM remodelling (ECM proteolysis and BM assembly) and cell exploratoriness significantly modify vascular network structure (see Fig 13 and S2 Fig). This is in agreement with experimental evidence of aberrant vessels with excessive branching in tumour-induced angiogenesis [86–88].

We calibrated and validated our model against *in vitro* experiments carried out over a time scale on which cell proliferation and cell death are negligible (only ≈ 5% of cells were undergoing mitosis in the observed time of ≈ 22.4 hours [2]). Whilst cell turnover is neglected in the present study, it will need to be incorporated in any future study that simulates larger vascular networks. To do so, we must first reduce the computational complexity of the model since, in its current implementation, the runtime of a single simulation is up to several hours (computational complexity increases with the number of cells in the system). One way to achieve this is to coarse-grain the subcellular model to a two-state (tip and stalk cell) Markovian system, omitting the dynamics of the intermediate variables [93]. We expect such an approach to reduce the computational complexity of the subcellular model, thus allowing us to run larger scale simulations. We also plan to scale up the model in order to obtain a continuum PDE limit description of it [81, 94–96]. This will further reduce the computational complexity of the simulations. Furthermore, a continuum PDE description of the model will allow us to perform a more systematic model calibration and sensitivity analysis of the model parameters.

To conclude, the model we have developed, with naturally emerging branching and EC chemotactic sensitivity to VEGF, allows us to investigate how changes in intracellular signalling and local cell environment influence cell mixing dynamics and to study the impact cell mixing has on the overall network structure. To our knowledge, it is the first attempt to quantify the cell mixing phenomenon in a theoretical model of angiogenesis. Since only individual cell trajectories are required for the computation of this statistic, it can also be extracted from experimental data. This, together with our predictions that cell mixing intensity is directly related to the vascular network structure, makes the mixing measure a potential marker for pathological

angiogenesis. Furthermore, although we used a specific formulation for our subcellular model, the same modelling approach can be applied to a more/less detailed system (e.g. recent works [97, 98]) as long as it reproduces typical phenotype patterning of cells within the vasculature. This flexibility allows us to use our model to test various experimental hypotheses or make predictions, for example, regarding pathological network formation.

## Supporting information

**S1 Appendix. Computational simulations.** The appendix provides details on setups of our numerical simulations and brief discussion on their computational implementation.
(PDF)

**S1 Fig. Examples of steady state patterns of the VEGFR-Delta-Notch subcellular model for different interaction radii.** Final steady state patterns established during single stochastic simulations of the system described by the kinetic reactions outlined in Fig 3D for a uniform hexagonal lattice of $10 \times 12$ voxels. **(A)** $R_s = 1.0h$, **(B)** $R_s = 2.0h$, **(C)** $R_s = 3.0h$ and the rest of the parameter values as in S1 Table.
(TIF)

**S2 Fig. Sensitivity analysis: Orientation vs. anterograde cell proportion.** We performed simulations of our model by varying one of the parameters of the cellular and tissue scales at a time by a fixed per cent and keeping default values for the rest of the parameters (as in S2 Table). Each parameter was varied by $\pm0.1\%$, $\pm1\%$, $\pm5\%$, $\pm10\%$, $\pm15\%$ and $\pm20\%$. For each numerical experiment, several quantitative metrics were computed. The results are represented as scatter plots of mean cell trajectory orientation vs. mean anterograde cell proportion with colouring indicating mean **(A)** cell number; **(B)** vascular network area; **(C)** vessel segment length; **(D)** number of branching points per 100 $\mu m^2$ of vascular network area and **(E)** displacements. On these plots, dashed magenta lines indicate the point corresponding to the default parameter values; magenta highlights the region of the main point clustering. The grey-coloured outlier region corresponds to vascular networks with less persistent, twisted vessels, whereas the brown outlier region is characterised by longer straight vessel segments. Variations of the parameters that push the system towards one of the outlier regions are indicated on each plot. Panel **(F)** provides a general summary of these results. Simulation setup as in **Setup 1**, S3 Table, with $T_{max} = 2.5$. The results are averaged over 100 realisations. The subcellular parameters were fixed at their default values in all experiments (see S1 Table).
(TIF)

**S3 Fig. Individual simulations of vascular networks generated by VEGFR2$^{+/-}$ mutant cells.** Final configurations of simulated vascular networks of VEGFR2$^{+/-}$ mutant cells growing in uniform concentration of VEGF = 5 ng/ml, plot **(A)**, and VEGF = 50 ng/ml, plot **(B)**. The leftmost panels show the amount of Delta, ***D***. Higher values (green colour) correspond to tip cell phenotype, low values (red colour)—to stalk. On these plots arrows correspond to the orientation landscape configuration, ***l***. The central panels indicate the final concentration of the ECM, ***c***. The rightmost panels—final distribution of the mean polarity angle, ***μ***, variable. Numerical simulations were performed using **Setup 1** from S4 Table and $T_{max} = 2.5$. Parameter values are listed in S1 and S2 Tables for subcellular and cellular/tissue scales, respectively, except of those changed for VEGFR2$^{+/-}$ mutant cells (see S1 Appendix).
(TIF)

**S4 Fig. Individual simulations of vascular networks generated by VEGFR1$^{+/-}$ mutant cells.** Final configurations of simulated vascular networks of VEGFR1$^{+/-}$ mutant cells growing in

uniform concentration of VEGF = 5 ng/ml, plot (**A**), and VEGF = 50 ng/ml, plot (**B**). The leftmost panels show the amount of Delta, **D**. Higher values (green colour) correspond to tip cell phenotype, low values (red colour)—to stalk. On these plots arrows correspond to the orientation landscape configuration, **l**. The central panels indicate the final concentration of the ECM, **c**. The rightmost panels—final distribution of the mean polarity angle, **μ**, variable. Numerical simulations were performed using **Setup 1** from S4 Table and $T_{max}$ = 2.5. Parameter values are listed in S1 and S2 Tables for subcellular and cellular/tissue scales, respectively, except of those changed for VEGFR1$^{+/-}$ mutant cells (see S1 Appendix).
(TIF)

**S5 Fig. Individual simulations of vascular networks generated by VEGFR2$^{+/-}$ mutant cells treated with DAPT.** Final configurations of simulated vascular networks of VEGFR2$^{+/-}$ mutant cells treated with DAPT growing in uniform concentration of VEGF = 5 ng/ml, plot (**A**), and VEGF = 50 ng/ml, plot (**B**). The leftmost panels show the amount of Delta, **D**. Higher values (green colour) correspond to tip cell phenotype, low values (red colour)—to stalk. On these plots arrows correspond to the orientation landscape configuration, **l**. The central panels indicate the final concentration of the ECM, **c**. The rightmost panels—final distribution of the mean polarity angle, **μ**, variable. Numerical simulations were performed using **Setup 1** from S4 Table and $T_{max}$ = 2.5. Parameter values are listed in S1 and S2 Tables for subcellular and cellular/tissue scales, respectively, except of those changed for VEGFR2$^{+/-}$ mutant cells and DAPT treatment (see S1 Appendix).
(TIF)

**S6 Fig. Individual simulations of vascular networks generated by VEGFR1$^{+/-}$ mutant cells treated with DAPT.** Final configurations of simulated vascular networks of VEGFR1$^{+/-}$ mutant cells treated with DAPT growing in uniform concentration of VEGF = 5 ng/ml, plot (**A**), and VEGF = 50 ng/ml, plot (**B**). The leftmost panels show the amount of Delta, **D**. Higher values (green colour) correspond to tip cell phenotype, low values (red colour)—to stalk. On these plots arrows correspond to the orientation landscape configuration, **l**. The central panels indicate the final concentration of the ECM, **c**. The rightmost panels—final distribution of the mean polarity angle, **μ**, variable. Numerical simulations were performed using **Setup 1** from S4 Table and $T_{max}$ = 2.5. Parameter values are listed in S1 and S2 Tables for subcellular and cellular/tissue scales, respectively, except of those changed for VEGFR1$^{+/-}$ mutant cells and DAPT treatment (see S1 Appendix).
(TIF)

**S7 Fig. Quantification of vascular network structure for WT and mutant cells at VEGF = 5 and 50 ng/ml.** (**A**) Number of vessel segments. (**B**) Vessel segment length ($\mu m$). (**C**) Vascular network area ($\mu m^2$) at the end of the numerical simulation. (**D**) Number of branching points per 100 $\mu m^2$ of vascular network area. Details of definitions of these metrics can be found in S1 Text. In each boxplot, the central line indicates the median, and the horizontal edges of the box represent the 25$^{th}$ and 75$^{th}$ percentiles (for the bottom and top edges, respectively). The outliers are indicated by red cross symbols. Numerical simulation setup used is **Setup 1** from S4 Table with final simulation time $T_{max}$ = 2.5. Parameter values are listed in S1 and S2 Tables for subcellular and cellular/tissue scales, respectively, except of those changed for mutant cells (see S1 Appendix). Results are averaged over 100 realisations for each experimental scenario.
(TIF)

**S8 Fig. Temporal evolution of mixing measure and tip cell proportion in simulated vascular networks.** Left column plots show the mixing measure, $M(t)$, as a function of time (the mean value is indicated by a thick line and standard deviation is shown by a band with

corresponding colour). Right column plots demonstrate the evolution of tip cell proportion. Simulations were done for networks formed by **(A)** WT cells; **(B)** VEGFR2$^{+/-}$ mutant cells; and **(C)** VEGFR1$^{+/-}$ mutant cells. Numerical simulation setup used is **Setup 1** from S4 Table with final simulation time $T_{max}$ = 2.5. Parameter values are listed in S1 and S2 Tables for sub-cellular and cellular/tissue scales, respectively, except of those changed for mutant cells (see S1 Appendix). Results are averaged over 100 realisations for each experimental scenario.
(TIF)

**S9 Fig. Mixing measure steady state for VEGF = 0 ng/ml.** Plots of mixing measure, $M(t)$, as a function of time for WT, VEGFR2$^{+/-}$ and VEGFR1$^{+/-}$ mutant cells for VEGF = 0 ng/ml (the mean value is indicated by a thick line and standard deviation is shown by a band of the corresponding colour). At this concentration of external VEGF, there is no effective sprout elongation, thus cells perform proteolysis-free random shuffling within already existing sprouts [52]. This leads to a steady state of the mixing measure for all cell lineages. Mean values are 0.39, 0.39 and 0.38 for WT, VEGFR2$^{+/-}$ and VEGFR1$^{+/-}$ cells, respectively. Numerical simulations were performed using **Setup 1** from S4 Table and $T_{max}$ = 2.5. Parameter values are listed in S1 and S2 Tables for subcellular and cellular/tissue scales, respectively, except of those changed for mutant cells (see S1 Appendix).
(TIF)

**S1 Table. Baseline parameter values for the VEGF-Delta-Notch subcellular model.** Description and reference values used in simulations of the subcellular VEGF-Delta-Notch signalling.
(PDF)

**S2 Table. Parameter values of the cellular and tissue scales used in our simulations.**
(PDF)

**S3 Table. Initial conditions for numerical simulations.** Here $\mathcal{I}$ is the set of all voxels; $\mathcal{S}$ is the set of all possible migration directions. DUnif[$a$, $b$] is a discrete uniform distribution over all integer numbers lying within the interval [$a$, $b$]; Unif[$a$, $b$] is the uniform distribution on the interval [$a$, $b$]. Baseline gene expression parameters for the VEGF-Delta-Notch signalling are listed in S1 Table. $\Delta_{init}$ = 1.0 for all numerical simulations (this value, as, in general, for the value of the OL variable, is non-dimensional). The fluctuation parameter, $\xi$, is set to 0.1 in all numerical simulations. The exact values for $c_{init}$ and $m_{init}$ are given for each numerical experiment in S4 Table, as well as the set of initial cell positions, $\mathcal{I}_{init}$. For the description of model variables see Table 1 in the main text.
(PDF)

**S4 Table. Setups of simulation experiments.** For each setup of numerical simulation we specify the lattice dimensions, $N_I^x$ and $N_I^y$; the set of indices corresponding to the vascular plexus, $\mathcal{I}_{VP}$; the initial cell nuclei positions, $\mathcal{I}_{init}$; the initial polarisation direction, $s_{init}$; the initial ECM and BM concentrations, $c_{init}$ and $m_{init}$, respectively; the VEGF distribution over the lattice, $V$; and cell line used in simulations.
(PDF)

**S1 Movie. An example of an individual vascular network generated by wild type ECs during simulation of our model with uniform VEGF = 50 ng/ml.** The leftmost panel shows the concentration of Delta, $D$. The colour bar indicates level of Delta, $D$, (green colour corresponds to tip cells, red—to stalk cells). Arrows indicate the configuration of the orientation landscape, $l$. The central panel indicates the concentration of the ECM, $c$. The rightmost panel —the polarity angle, $\mu$, variable. A circular colour bar indicates the value of $\mu$. The simulation was performed using **Setup 1** from S4 Table with final simulation time, $T_{max}$ = 2.5. Parameter

values are listed in S1 and S2 Tables for subcellular and cellular/tissue scales, respectively.
(MP4)

**S2 Movie. Cell migration from a cell bead in substrates of different collagen density.** Single realisations of angiogenic sprouting from a cell bead in substrates of different collagen densities (reproducing the results of the polarisation experiment in [38]). Maximum collagen density **(A)** $c_{max}$ = 0.1, **(B)** $c_{max}$ = 1.0, **(C)** $c_{max}$ = 1.7, **(D)** $c_{max}$ = 3.0. The VEGF linear gradient starts with 0 ng/ml at $y$ = 0 and increases up to 5 ng/ml at $y$ = 125 $\mu m$. Central bead initial and basement membrane conditions, $\mathcal{I}_{BM} = \mathcal{I}_{init}$, are outlined by a black thick line on each plot. Colour bars indicate the level of Delta ligand. The simulations were performed using **Setup 3** from S4 Table. Parameter values are listed in S1 and S2 Tables for subcellular and cellular/tissue scales, respectively.
(MP4)

**S3 Movie. Single realisations of cells shuffling within a linear sprout when two given cell lines are mixed 1:1 (50% to 50%).** The cell lines used in each realization are indicated in the titles. In the top row, no treatment with DAPT inhibitor was applied to cells; in the bottom row, all ECs were treated with DAPT. The leading edge corresponds to two rightmost voxels of each sprout. The colour bar for Delta level of the WT goes from red colour (stalk cell) to green (tip cell), whereas for the mutant cells the bar goes from purple colour (stalk cell) to yellow (tip cell). The simulations were performed using **Setup 4** from S4 Table. Parameter values are listed in S1 and S2 Tables for subcellular and cellular/tissue scales, respectively, except for the changed parameters for the mutant cells listed in S1 Appendix. Final simulation time, $T_{max}$ = 50.0.
(MP4)

**S4 Movie. Examples of an individual vascular networks generated by wild type and mutant (VEGFR2$^{+/-}$ and VEGFR1$^{+/-}$) ECs during simulation of our model with uniform VEGF = 5 ng/ml.** The cell line is indicated in the title of each panel. The colour bar indicates level of Delta, $D$, (green colour corresponds to tip cells, red—to stalk cells). Arrows indicate the configuration of the orientation landscape, $l$. Numerical simulation was performed using **Setup 1** from S4 Table with final simulation time, $T_{max}$ = 2.5. Parameter values are listed in S1 and S2 Tables for subcellular and cellular/tissue scales, respectively, except for the changed parameters for the mutant cells listed in S1 Appendix.
(MP4)

**S1 Text. Supplementary material.** The file contains a more detailed description of the subcellular VEGF-Delta-Notch model, metric definitions and algorithms for their implementation, further specifications on simulation of our model and sensitivity analysis of model parameters.
(PDF)

## Author Contributions

**Conceptualization:** Daria Stepanova, Helen M. Byrne, Philip K. Maini, Tomás Alarcón.

**Investigation:** Daria Stepanova, Helen M. Byrne, Philip K. Maini, Tomás Alarcón.

**Methodology:** Daria Stepanova, Helen M. Byrne, Philip K. Maini, Tomás Alarcón.

**Supervision:** Helen M. Byrne, Philip K. Maini, Tomás Alarcón.

**Writing – original draft:** Daria Stepanova, Helen M. Byrne, Philip K. Maini, Tomás Alarcón.

**Writing – review & editing:** Daria Stepanova, Helen M. Byrne, Philip K. Maini, Tomás Alarcón.

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
