## [Decision Letter · Decision Letter 0]

10 Aug 2020

Dear Mrs. Stepanova,

Thank you very much for submitting your manuscript "A multiscale model of complex endothelial cell dynamics in early angiogenesis" for consideration at PLOS Computational Biology.

As with all papers reviewed by the journal, your manuscript was reviewed by members of the editorial board and by several independent reviewers. In light of the reviews (below this email), we would like to invite the resubmission of a significantly-revised version that takes into account the reviewers' comments.

In particular, based upon the reviewers' comments, I would ask that a more extensive parameter sensitivity analysis  is performed to strengthen the comparison with published experimental work and to arrive at a number of  model predictions that can be experimentally tested in the future. Suggestions of recently published experimental insights are given in the reviews. Also reviewers ask for more in-depth quantitative analysis of the results. Please address these and the other comments in your revisions. 

We cannot make any decision about publication until we have seen the revised manuscript and your response to the reviewers' comments. Your revised manuscript is also likely to be sent to reviewers for further evaluation.

Sincerely,

Roeland M.H. Merks, Ph.D

Associate Editor

PLOS Computational Biology

Mark Alber

Deputy Editor

PLOS Computational Biology

Reviewer's Responses to Questions

**Comments to the Authors:**

Reviewer #1: In "A multi-scale model of complex endothelial cell dynamics in early angiogenesis" the authors present a mathematical model of the early stages of sprouting angiogenesis particularly tailored to describe the dynamics of the endothelial cell mixing within a growing sprout. Though fairly abstract (since it does not reproduce the shape/diameter of the growing vessel, and forces specific angles for bifurcations, see below), it includes several of the main mechanism driving vessel sprouting: notch signalling, ECM remodelling and endothelial cell polarisation. In particular, the descriptions of the notch signalling mechanism and ECM remodelling are novel, simple and thorough. The manuscript is therefore worthy of publication in PLoS Computational Biology once the following items are addressed:

1: In line 180 it is mentioned that "...actin filaments close to the membrane... can bind more VEGF and ECM proteins." However, actin filaments do not bind VEGF and ECM proteins directly.

2: In lines 188/189 and then on the results section (lines 569 onwards) the authors mention the "brush-border" effect. The authors should compare directly the model results with experiments on how branch density increases with VEGF concentration. In particular experiments of our collaborators indicate that there is a saturation VEGF concentration above which the number of sprouts starts decreasing. Is the model able to reproduce this VEGF concentration saturation? Here, if I understood correctly, the proposed mechanism is: more VEGF leads to more tip cells and Delta expression, which lead to more ECM degradation and more branches. So the question is what is the behaviour of the simulated Notch-Delta stochastic model and ECM proteolysis at high VEGDF concentrations.

3: The Notch dynamics is modelled though a master equation simulated via a NSV algorithm. Can the authors refer to experimental evidence of stochastic Notch-Dll4 phenotype switching? Bentley et al have done extensive experiments on the time dependence of Delta-4 expression in endothelial cells and find that the oscillations in expression are periodic and can be explained by the coordinated response of groups of endothelial cells to high levels of VEGF. How do the results of the simulations depend on the stochastic nature of Notch-Delta dynamics? Also, how does the presence of ECs intermediate states (neither stalk not tip cell, predicted in Bentley et al models) would affect the model predictions for EC mixing?

4: ECs are polarised and this polarisation determines EC movement in the model. However, cell contacts are not influenced by the polarisation direction, and are determined instead by the other cells within a circular neighbourhood. Have the authors explored the possibility of an elliptical neighbourhood aligned with the polarisation direction to determine Notch-Delta interactions? How would this alteration affect the model results?

5: Explain in greater detail the reasoning behind the terms in equation (9).

6: The branching angles in the model are strongly influenced by the lattice. Once the new tip cell starts exploring other directions, its movement is locked at a 60 degree angle relative to the initial vessel since it has no other option. If there were no lattice I would expect that the new tip cell could in principle bifurcate with a wide range of possible angles relative to the initial vessel. In particular the new vessel could grow almost parallel to the original vessel; I would even expect that angles close to zero would be the most probable. How can the model become lattice-free and achieve more realistic vessel geometries?

7: The model has a very large number of parameters. The authors calibrate the model choosing a combination of parameters by comparison to experiments. I expect that there are other possible choices of parameters that will lead to a similar performance of the model. In which order the authors chose the parameters? Which ones were set to fixed values and which ones were fit by the experimental results?

8: In page 28 the authors show that the ratio of cells undergoing anterograde movement relative to retrograde movement increases with VEGF levels. The authors should better describe the mechanisms underlying this finding. How does the probability for the cell to undergo retrograde movement depends on the microenvironment (ECM concentration and alignment, basement membrane, VEGF...) and on the relative position and phenotype of neighbouring cells?

9: In line 741 the authors mention that "The vascular networks generated by our model are in good 741 agreement with experimental observations". Well, not quite: there are several features that are in good agreement but the general network aspect is quite different from experimental results (notably vessel structure depends dramatically on the conditions in which vessels grow, 2D vs 3D, retina vs heart, ex-vivo vs in-vivo, etc..). I suggest the authors to soften the sentence.

Reviewer #2: The authors have developed a 2-D multiscale model for angiogenesis and integrated cellular signaling (subcellular) and mechanical (cell-ECM interactions-cellular/tissue level) elegantly. They also investigate cell mixing and make specific predictions about some experimental scenarios. While the manuscript is largely well-done, it falls short of making a stronger connect with the existing literature and interpreting their results more carefully:

1. The authors have incorporated the aspect of cis-inhibition in their model, but no robust experimental evidence exists for cis-inhibition in angiogenesis. How do the results presented change or are expected to change if cis-inhibition is weakened or removed? The role of cis-inhibition in Notch-Delta signaling has been well-explored in terms of cell-fate decisions (Sprinzak et al. Nature 2010, Jolly et al. New J Physics 2015).

2. The authors discuss about cell mixing/rearrangements in sprouting angiogenesis. Does this mixing continue in homeostatic conditions? If yes, what is the need and implication of such mixing? If not, what mechanisms lead to stopping this behavior? Is this mixing a completely cell-autonomous decision or also influenced by biochemical and/or biomechanical crosstalk with the neighbors? Recent evidence about cell rearrangements has been reported in collective cell migration (Zhang et al. PNAS 2019). The authors should discuss similarities and differences with the mechanisms at play there.

3. The authors have implemented a 2D framework of Notch signaling coupled with VEGF signaling. They should briefly discuss how similar or different their frameworks is from other attempts to simulate Notch signaling in 2D and its coupling with cell-fate decisions such as EMT (Boareto et al. J R Soc Interface 2016).

4. The authors mention the asymptotic value of mixing measure as 0.385. What is the physical interpretation of this value? How does this value depend on model parameters?

5. The authors have not performed a detailed sensitivity analysis. Given their framework, one of the approaches in enabling more confidence about the robustness of their predictions can be performing some principal component analysis to map combinations of input parameters to specific outputs (Fig 2; Pramanik et al. bioRxiv 2020, 041632).

6. The authors mention about a “sweet spot” of collagen density. They should discuss their results in the context of a recent study showing optimality of cell invasion at intermediate matrix stiffness (Ahmadzadeh et al. PNAS 2017).

7. Can the authors make an effort to connect their predictions to latest observations about dynamics of angiogenesis, such as in Kang et al. PNAS 2019, Antfolk et al. PNAS 2017?

Reviewer #3: To the Authors:

In this paper, the authors propose an experimental data-driven hybrid two-dimensional multiscale model that reproduces complex endothelial cell (EC) dynamics, including cell rearrangements, via heterogeneous response to their microenvironments and cell-cell interactions in early angiogenesis. The model also well recapitulated characteristic features of angiogenesis such as branching, chemotactic sensitivity and brush border effect that emerge naturally from the gene expression patterns of individual cells. They finally used it to predict how the structure of the vascular network changes as the baseline gene expression pattern of VEGF-Delta-Notch signaling pathway and the composition of the extracellular environment vary, using the model. Especially, they addressed an unresolved important question regarding impacts of EC rearrangements on the vascular network structure by introducing an index, mixing measure that quantifies cell mixing as the vascular network grows. Then, they found a direct correlation between EC rearrangement and the generic features of the vascular branching pattern, in which lower EC rearrangement can lead to an imbalance between branching and sprout elongation. The proposed model is novel and seems quite useful to systematically dissect molecular and cellular mechanisms underlying angiogenic morphogenesis via model predictions and experimental validations, especially functional involvement of cell rearrangement, which can lead to scientific progress in the research field. Additionally, in whole, study signs are well organized and the paper is well written. However, some following points need to be addressed to make it more valuable.

1. The reviewer recommends to rearrange the composition of the paper (text and figures) to more highlight a part of “Model predictions”, which will more interest the readers, especially biologists. The author may wish to relatively reduce the parts of model construction and validations to more appeal model prediction.

2. The authors need to quantify the vascular morphology they simulated using some indices i.e. branching points and branch lengths to more efficiently analyze causative relationships between cell dynamics and morphological changes. This seems important to emphasize one of the merits of the model.

3. The authors should make some efforts to evaluate their model prediction data by comparing biological data in the previous reports, if present. The reviewer understands that complete validation of their model prediction might be difficult at this moment, but they can discuss the validity using the similar biological data previously reported, which will provide important information to the readers.

4. Notation for VEGF receptor mutant cells, R2+ an R1+, is quite confusing to the reviewer and maybe the readers including biologists, because generally VEGFR2 wild type, heterozygote and homozygote cells are represented as VEGFR2+/+, VEGFR2+/- and VEGFR2-/-, respectively. The reviewer recommends to change the notation to others.

**Have all data underlying the figures and results presented in the manuscript been provided?**

Reviewer #1: Yes

Reviewer #2: Yes

Reviewer #3: Yes

PLOS authors have the option to publish the peer review history of their article (what does this mean?). If published, this will include your full peer review and any attached files.

Reviewer #1: No

Reviewer #2: No

Reviewer #3: **Yes: **Koichi Nishiyama
---

## [Decision Letter · Decision Letter 1]

19 Nov 2020

Dear Mrs. Stepanova,

We are pleased to inform you that your manuscript 'A multiscale model of complex endothelial cell dynamics in early angiogenesis' has been provisionally accepted for publication in PLOS Computational Biology.

Best regards,

Roeland M.H. Merks, Ph.D

Associate Editor

PLOS Computational Biology

Mark Alber

Deputy Editor

PLOS Computational Biology

Reviewer's Responses to Questions

**Comments to the Authors:**

Reviewer #1: The authors have answered all my queries.

Reviewer #2: The authors have now addressed my comments satisfactorily.

Reviewer #3: The authors sufficiently revised the manuscript according to the reviewer's comments, which satisfies the reviewer.

**Have all data underlying the figures and results presented in the manuscript been provided?**

Reviewer #1: None

Reviewer #2: None

Reviewer #3: Yes

PLOS authors have the option to publish the peer review history of their article (what does this mean?). If published, this will include your full peer review and any attached files.

Reviewer #1: No

Reviewer #2: No

Reviewer #3: **Yes: **Koichi Nishiyama

---

## [Editor Report · Acceptance letter]

22 Dec 2020

PCOMPBIOL-D-20-00995R1 

A multiscale model of complex endothelial cell dynamics in early angiogenesis

Dear Dr Stepanova,

I am pleased to inform you that your manuscript has been formally accepted for publication in PLOS Computational Biology. Your manuscript is now with our production department and you will be notified of the publication date in due course.

With kind regards,

Jutka Oroszlan
